# The structure of the complete extracellular bacterial flagellum reveals the mechanism of flagellin incorporation

Rosa Einenkel[1,11], Kailin Qin[2,11], Julia Schmidt[1], Natalie S. Al-Otaibi[2,6], Daniel Mann [3], Tina Drobnič [4,7], Eli J. Cohen[4], Nayim Gonzalez-Rodriguez [4,8], Jane Harrowell [4,9], Elena Shmakova [4,10], Morgan Beeby [4], Marc Erhardt [1,5] ✉ & Julien R. C. Bergeron [2] ✉

The bacterial flagellum is essential for motility, adhesion and colonization in pathogens such as *Salmonella enterica* and *Campylobacter jejuni*. Its extracellular structure comprises the hook, hook–filament junction, filament and filament cap. Native structures of the hook–filament junction and the cap are lacking, and molecular mechanisms of cap-mediated filament assembly are largely uncharacterized. Here we use cryo-electron microscopy to resolve structures of the complete *Salmonella* extracellular flagellum including the pentameric FliD cap complex (3.7 Å) and the FlgKL hook–filament junction (2.9 Å), as well as the *Campylobacter* extracellular flagellum before filament assembly (6.5 Å). This, coupled with structure-guided mutagenesis and functional assays, reveals intermediates of filament assembly, showing that FliD cap protein terminal domain movement and clockwise rotation enable flagellin incorporation and stabilization of the filament. We show that the hook–filament junction acts as a buffer, preventing transfer of mechanical stress to the filament, and reveal the structural basis for the initiation of filament assembly. Collectively, this study provides comprehensive insights into flagellum assembly and how flagellin incorporation is coupled with its secretion.

The flagellum is the most prominent extracellular structure in bacteria, with a molecular weight in the hundreds of megadaltons. It allows them to move within their environment through the rapid rotation of its propeller-like filament. In many human pathogens, including the prominent gastrointestinal pathogens *Salmonella enterica* and *Campylobacter jejuni*, the flagellum also plays an important role in infection, because of its ability to promote adhesion and colonization[1]. Structurally, the flagellum can be divided into three major components:

basal body, hook and filament (Fig. 1a). The hook, a helical assembly of hundreds of subunits of FlgE, functions as a universal joint that connects the extracellular filament to the membrane-embedded basal body[2,3]. The flagellar filament, a multi-micron structure comprising tens of thousands of subunits of a single protein, the flagellin, facilitates bacterial motility through its rotation[4]. Structural studies have revealed that the filament is a superhelical assembly consisting of 11 protofilaments[5–8]. A ring-like structure formed by the proteins

[1]Institute of Biology, Humboldt-Universität zu Berlin, Berlin, Germany. [2]Randall Centre for Cell and Molecular Biophysics, King's College London, London, UK. [3]Forschungszentrum Jülich GmbH, Jülich, Germany. [4]Imperial College London, London, UK. [5]Max Planck Unit for the Science of Pathogens, Berlin, Germany. [6]Present address: School of Natural Sciences, Birkbeck, University of London, London, UK. [7]Present address: MRC laboratory of Molecular Biology, Cambridge, UK. [8]Present address: Spanish National Cancer Research Center, Madrid, Spain. [9]Present address: CyanoCapture, Oxford, UK. [10]Present address: Max Planck Institute of Biochemistry, Munich, Germany. [11]These authors contributed equally: Rosa Einenkel, Kailin Qin. ✉e-mail: marc.erhardt@hu-berlin.de; julien.bergeron@kcl.ac.uk

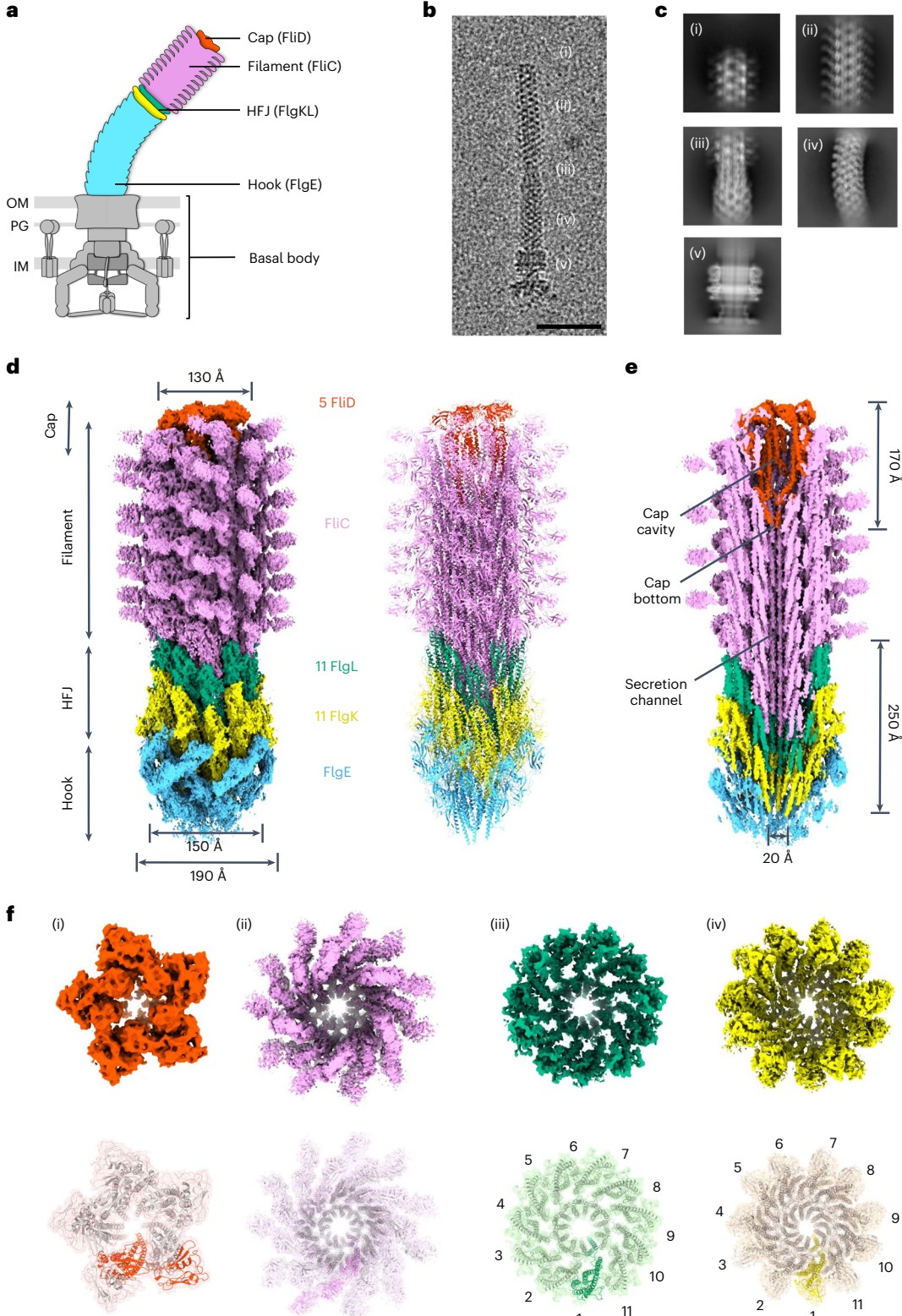

**Fig. 1 | The complete structure of the extracellular flagellum. a**, Schematic representation of a Gram-negative bacterial flagellum. IM, inner membrane; PG, peptidoglycan layer; OM, outer membrane. **b**, Isolated area of a representative micrograph showing an intact flagellum: (i) cap region, (ii) filament, (iii) HFJ, (iv) hook and (v) basal body. Scale bar, 50 nm. **c**, Representative 2D class averages of the (i) cap region, (ii) filament, (iii) HFJ, (iv) hook and (v) basal body. **d**, Composite map of the extracellular flagellar complex from the hook to the cap. The map of the flagellar tip and the map of the HFJ are refined individually and connected through multiple layers of flagellin. Atomic model of the entire extracellular flagellum (Protein Data Bank (PDB): 9GNZ and 9GO6). **e**, Cross-section view of the composite map, with dimension of the FliD cap and HFJ indicated. **f**, Top view of each section of the extracellular flagellum. Maps (top) and models fitted into the map, shown in the same colour (bottom), with one subunit for each protein highlighted, are shown. (i) FliD pentamer. (ii) FliC filament. (iii) FlgL layer with its stoichiometry in the HFJ. (iv) FlgK layer with its stoichiometry in the HFJ.

FlgK and FlgL, termed the hook–filament junction (HFJ), forms the connection between the flexible hook and the rigid filament[9]. The structures of FlgK and FlgL have been previously reported, but only in their monomeric form[10–14]. To assemble a filament, flagellin monomers are secreted through the flagellar basal body and hook, and polymerize at the distal end. The insertion of flagellin into the growing filament is mediated by the filament cap complex, which assembles atop the HFJ and is composed of the protein FliD[15–19]. On the basis of the structure of the filament cap complex in isolation, we previously proposed that FliD forms a stable pentameric complex that rotates to enable the incorporation of new flagellin subunits into the respective protofilaments[20]. How the filament self-assembles, however, has remained elusive. Particularly, the molecular mechanism of the filament cap in facilitating the assembly of the filament and the function of the HFJ as a template for correct filament assembly are not yet understood.

Here we report the structure of the filament cap complex in its native environment, assembled at the distal end of the flagellar filament. Critically, we were able to determine this structure at various stages of flagellin incorporation into the filament, allowing us to identify the molecular steps involved in this process. Using mutagenesis and functional assays, we show that the FliD terminal domains are essential for flagellin folding and incorporation into the filament. Furthermore, we report the structure of the intact HFJ within a fully assembled flagellar filament, experimentally confirming the proposed 11:11 stoichiometry of the FlgKL proteins and revealing the molecular details of its gasket-like role in isolating the filament from the hook. Structure-guided mutagenesis, which destabilizes the FlgKL interface, highlights the importance of FlgKL intermolecular interactions for structural integrity and proper function of the flagellar apparatus. Finally, we report the structure of the cap assembled on the HFJ, corresponding to the state of the complex before filament assembly, which reveals the structural changes that occur in the cap upon the initiation of filament formation. Collectively, these results reveal a detailed model for cap-mediated flagellin insertion, illustrating how flagellin incorporation is coupled with secretion.

## Results

### The complete structure of the extracellular flagellum

We previously reported the isolated flagellum cap complex as a pentameric structure with five-fold symmetry[20]. To determine its native structure, we applied cryogenic electron tomography (cryo-ET) on intact *S. enterica* cells, obtaining 68 tomograms with 252 flagellar ends for subtomogram averaging, resulting in a ~25-Å map (Extended Data Fig. 1a–c, Supplementary Table 1 and Supplementary Video 1). In this map, a visible seam indicated asymmetric positioning of the FliD subunits within the filament (Extended Data Fig. 1c). This demonstrated structural differences between the native state and the structure of the isolated cap complex[20]. Due to limited resolution, detailed FliD–flagellin interactions remained unresolved, prompting us to use single-particle cryo-EM.

To this end, we genetically modified *S. enterica* to generate flagellum complexes with short filaments. To increase flagellum yield during purification, we exchanged the native promoter of the flagellar master regulator FlhDC with the strong synthetic constitutive promoter $P_{proB}$, generating hyperflagellated cells[21,22]. To control filament length, we locked the cells in the production of the flagellin FliC and exchanged the native *fliC* promoter with an inducible $P_{tetA}$ promoter. A 30-min induction produced short filaments and prevented filament breakage and cap loss during purification (Extended Data Fig. 1d). Single-cell tracking and quantification of filament length revealed that these short filaments failed to promote motility, remaining shorter than the previously reported minimal length of ~2.5 µm required for motility (Extended Data Fig. 1e,f)[23]. Flagella were purified as described previously[24–26], yielding intact complexes encompassing basal body, hook, junction, filament and cap (Fig. 1b,c).

Cryo-EM analysis of purified flagella with short filaments (Extended Data Fig. 2) allowed us to obtain independent maps for the cap and HFJ, to 3.7 Å and 2.9 Å resolution, respectively (Supplementary Table 1). An initial cap complex map at 3.3 Å did not resolve the D2 and D3 domains of one FliD subunit, indicating structural heterogeneity. Three-dimensional (3D) classification identified the different states of the cap, resulting in a final 3.7-Å map resolving all five FliD subunits (Extended Data Fig. 2). The final cap model contains 5 FliD subunits and 17 copies of FliC in the filament; the HFJ model contains 13 copies of FlgE, 11 copies of FlgK and FlgL, and 14 copies of FliC. Aligning these maps via overlapping FliC molecules yielded a composite structure of the distal FlgE hook, FlgKL HFJ, FliC filament and FliD cap (Fig. 1d–f and Supplementary Video 2). The cap measures ~130 Å in width and ~170 Å in length, with the D2–D3 plane tilted similarly to tomography observations (Fig. 1d,e and Fig. 2a, and Extended Data Fig. 1c). A cavity enclosed by D2–D3 and D0–D1 of the cap is consistent with early low-resolution data[17]. FlgK and FlgL assemble into the HFJ in individual layers (Fig. 1d–f). The HFJ comprises distinct layers of FlgK (~190 Å width) and FlgL (~150 Å width), extending ~250 Å overall. A total of 11 FlgL subunits form the distal HFJ layer beneath the filament, while 11 FlgK subunits form the proximal layer above the hook (Fig. 1d–f). The stoichiometry and arrangement of FlgKL are consistent with previous suggestions that were made based on structural modelling[10,12,13].

### Native cap structure and its interaction with the filament

The flagellar cap complex is composed of five FliD subunits assembled into a pentamer with an overall shape resembling an acorn. Consistent with our previous structure of the cap in isolation[20], the FliD monomer consists of four domains, termed D0 to D3. The D2–D3 domains assemble into a star-shaped plane and the D0–D1 helices form a plug-like structure beneath (Fig. 2a). The D2–D3 plane and D0–D1 helices surround a large cavity, much larger than that described previously[17], and the D0 domains form an almost closed constriction in the secretion channel (Fig. 2a). The cavity adopts an oval shape around the D1 domains, ~45 Å wide along the long axis and ~36 Å wide along the short axis (Fig. 2a(i)). The cavity is narrower across the D0 domain, ~27 Å (long axis) and ~15 Å (short axis) (Fig. 2a(ii)). The bottom of the cavity is formed by C-terminal D0 domains (hereafter referred to as D0-C), which form a constriction with a diameter of ~14 Å, continuous with the secretion channel in the filament (Fig. 2a(iii)). This constriction is narrower than that of the channel in the filament, which is 20 Å (Fig. 2a(iv)).

The D0–D1 domains of FliD are oriented vertically, while the D2–D3 domains are horizontal and point in the counterclockwise (CCW) direction (Fig. 2b,c). The D3 domain of one FliD stacks on the D2 domain of the adjacent FliD in the CCW direction (Fig. 2c). Owing to the varying heights of each FliD subunit, the D2–D3 plane is tilted, with angles between D0–D1 and D2–D3 of 84° (FliD 1), 81° (FliD 2), 56° (FliD 3), 73° (FliD 4) and 82° (FliD 5) (Fig. 2c).

All five FliD subunits show a pseudosymmetric arrangement, forming an asymmetric unit with three adjacent FliC molecules, labelled FliC 0, FliC +5 and FliC +6 (Fig. 2b). Using the protein, surfaces and assemblies service (PISA) at the European Bioinformatics Institute (EBI)[27,28], we analysed the interactions between FliD subunits and their adjacent FliC molecules. FliD subunits 1, 2, 3 and 5 form an extensive interface with FliC +5, with the interface being twice as large as that with FliC 0 or FliC +6. By contrast, FliD 4 interacts only with FliC +5 and FliC 0, but not with FliC +6 (Extended Data Table 1). Moreover, FliD 4 forms a substantially smaller interface with FliC +5. These unique features indicate that FliD 4 participates in the incorporation of a new FliC, which occurs in the gap between FliD 4 and FliC +6.

### Conformational changes to the cap upon filament assembly

As indicated above, we noticed that one FliD subunit is not well resolved in our initial map (Extended Data Fig. 2). We hypothesized that this FliD subunit is flexible and adopts different conformations. To verify

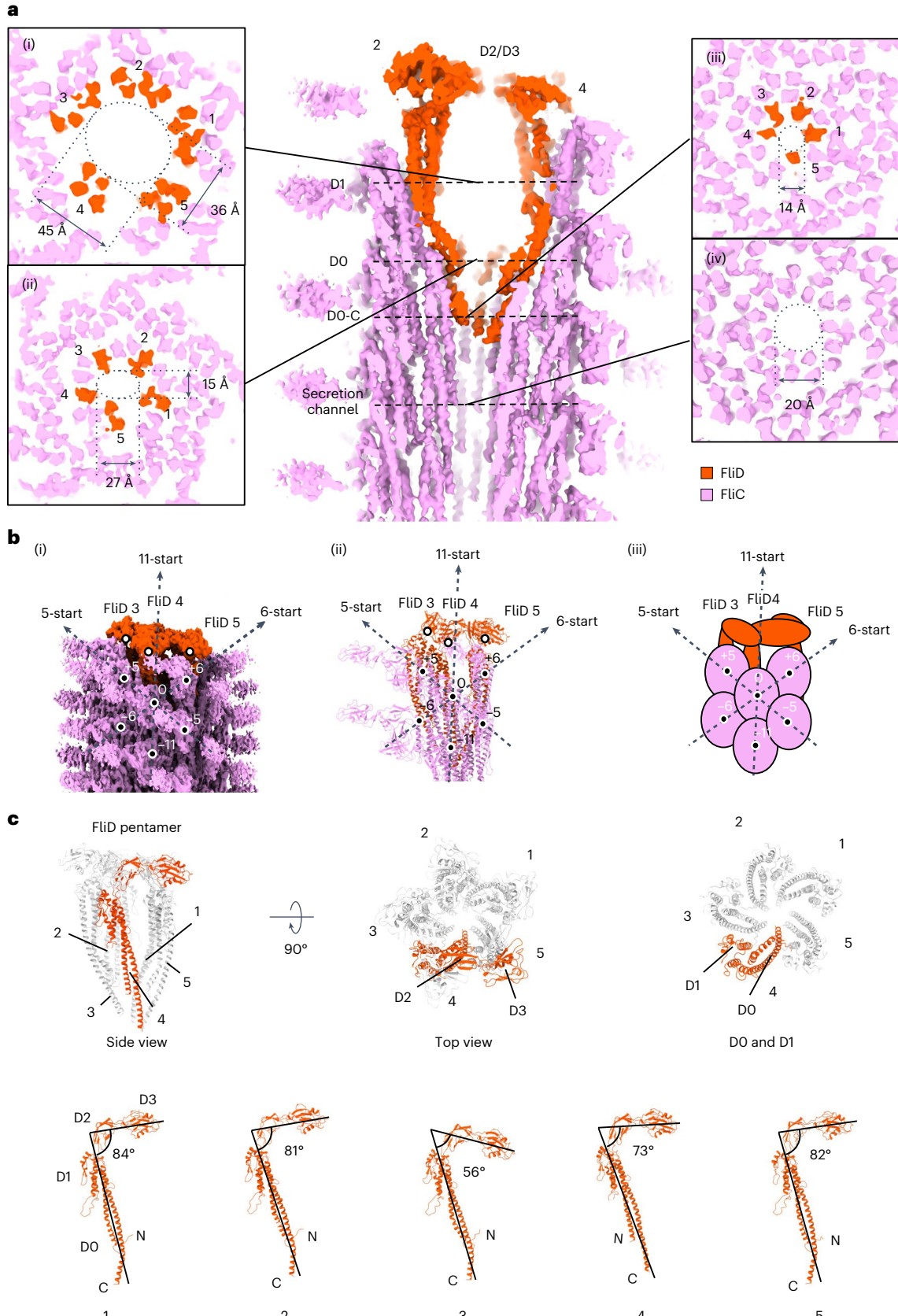

**Fig. 2 | Structure of the native cap complex and its interaction with the filament. a**, Cross-section of the FliD cap at the distal end of the filament and the dimension of the cavity. (i) Cross-section at D1 helices at the widest part of the cavity. (ii) Cross-section at D0 helices. (iii) Cross-section at D0-C helices at the bottom of the cavity. (iv) Cross-section of the FliC filament below the FliD cap.

**b**, Arrangement of FliD and FliC in the cap–filament complex. (i) Symmetry array of FliC in the filament and FliD position relative to FliC. (ii) Arrangement of subunits shown as the atomic model (PDB: 9GNZ). (iii) Schematic representation of FliC symmetry and relative positions of FliD and FliC. **c**, Overall atomic structure of the FliD cap and structural polymorphism among FliD subunits.

this, we performed 3D variability analysis (3DVA) in CryoSPARC[29]. This allowed us to obtain individual maps with assembly intermediates, whereby flagellin subunits are in the process of being incorporated, and FliD monomers adopt distinct conformations to that of the final map described above (Fig. 3 and Supplementary Video 3). Notably, we were able to reconstruct the molecular motion in the cap complex during filament assembly (Supplementary Video 4). During the incorporation of a FliC molecule, the D0–D1 domains of FliD 4 rise upwards ($\Delta = 25$ Å) and shift in the clockwise (CW) direction ($\theta = 18°$). The D2 domain of FliD 4 also rises, thereby causing the rise of the D3 domain of FliD 3, while the N-terminal loop of FliD 4 rises and turns 90° upwards (Supplementary Video 4). Finally, the D3 domain of FliD 4 rises, which is caused by the rise of the D2 domain of FliD 5, completing the cycle.

We also observed distinct intermediates of FliC folding upon incorporation into the filament (Supplementary Video 3 and Fig. 3a,b). Densities for the D0 and D1 helices of the N-terminus, which are close to the D0-C domain of FliD and the D1 domain of FliC 0, respectively, emerge initially (Fig. 3b). Subsequently, densities of helices in the C-terminal D0–D1 domains emerge, which are close to the N-terminal D0 domain (hereafter referred to as D0-N) of FliD and the D1 domain of FliC +6, respectively, and more density can be seen for the helices in D0-N and D1-N of FliC. Following this, helices in the D0 and D1 domains, the β-sheet and loops in the D1 domain, and the loop at the N-terminus are completed. At this stage, the density for the D2 domain begins to emerge. After D2 completion, the D3 domain starts to appear (Fig. 3b). Notably, before maturation of one FliC monomer is complete, incorporation of the next FliC molecule begins (Fig. 3b,c). Given the spatial patterns of domain appearance, we propose the following folding sequence for FliC: helices of D0-N and D1-N, helices of D0-C and D1-C, β-sheet and loop in D1 and N-loop, and D2 and D3 domains (Fig. 3d).

To validate these findings, particles of different 3DVA clusters and corresponding volumes were subjected to non-uniform refinement (Extended Data Fig. 3). Clusters 2 to 0 represent the first incorporation while clusters 19 to 8 represent the second FliC incorporation, which is consistent with the 3DVA results (Supplementary Video 3). Three-dimensional volumes were reconstructed using particles from clusters 12, 4, 15 and 1, respectively. The various FliC folding steps upon incorporation are also visualized and consistent with 3DVA frames in Fig. 3a, supporting the presence of the observed intermediates of FliC folding.

## FliD terminal regions mediate flagellin insertion

On the basis of our structural studies, we investigated the importance of FliD terminal domains in filament elongation in *S. enterica* using genetic engineering and functional assays. Our structural analysis revealed extensive interactions between FliD terminal regions and FliC domains D0–D1 (Fig. 4a,b and Extended Data Fig. 4). To test the relevance of these interactions, we introduced targeted serine (small, polar residue) and arginine (large, charged residue) substitutions at key interface residues into the native *fliD* locus and assessed their impact on motility and filament assembly.

Motility assays (Fig. 4c) showed that substituting conserved hydrophobic residues in the D0 domain of FliD to serine has no (V9S, F461S) or mild (L22S, F440S, L443S, M446S, L450S and Y456S) impact on swimming ability. Similarly, serine mutations in residues of the D1 domain involved in flagellin interactions result in mild (Y296S) or no (R319S) reduction of motility. Double mutation of the D1 domain residues (Y296S R319S) and substitution of Y296 with arginine result in a 10% decrease in motility.

However, double serine mutants of the D0 domain (V9S F440S, V9S L443S, V9S L450S, V9S F461S) show more pronounced motility defects (70–90% of wild-type (WT) motility). Furthermore, arginine substitutions in terminal residues result in significant motility reductions: F440R and L443R and the V9S F440R double mutant show

approximately 50% WT motility, while V9R and L22R retain about 80%. The most severe defect is observed in the V9S L443R double mutant, with 41% of WT motility. Overall, our data suggest that substitutions to small, polar residues are tolerated, while substitutions to larger, charged residues impair motility.

To confirm that the motility defects result from impaired filament assembly, we studied the flagellation patterns of the cap mutants using fluorescence microscopy (Fig. 4d–f and Extended Data Fig. 5a,b). We used a pulsed *flhDC* induction set-up (Extended Data Fig. 5e) to ensure synchronization of flagellum biosynthesis and visualized the filaments using a fluorophore-coupled maleimide dye (Fig. 4e and Extended Data Fig. 5a,b). The WT and mutant bacteria possess an average of four filaments per cell, indicating that the mutations do not affect the genetic regulation of flagellum biosynthesis (Fig. 4d). Importantly, filament length is significantly reduced in the analysed cap mutants with severely reduced motility (V9R and F440R), being on average 38% (V9R) or 52% (F440R) shorter than WT filaments (Fig. 4f). These results confirm that the reduced motility resulting from these mutations is due to impaired filament assembly.

In addition, secretion assays confirmed that cellular and secreted levels of FliD remained comparable between mutants and WT (Extended Data Fig. 5c), excluding impaired secretion or premature degradation. However, increased secretion of FliC in the mutants indicated inefficient flagellin incorporation (Extended Data Fig. 5d).

We next confirmed flagellin leakage during filament formation by separating cellular flagellin, flagellin attached to the cell body, detached filaments and leaked flagellin monomers using differential (ultra)centrifugation (Fig. 4g). In the WT, 17% of secreted FliC is found in the supernatant in monomeric form, whereas most of the flagellin is incorporated into the filament (Fig. 4h,i). By contrast, both the V9R and the F440R mutant show a significant increase in monomeric flagellin in the supernatant (31% and 35%, respectively), consistent with fluorescence microscopy and secretion assay results. In the $\Delta flgKL$ mutant, FliC is mostly present (62%) in the supernatant, confirming that this mutant does not permit filament assembly. We note that some level of cell-attached flagellin is detected in this mutant, probably owing to enhanced secretion and aggregation of FliC monomers. Collectively, our data show that the filament cap mutants incorporated nascent flagellin less efficiently, resulting in shorter filaments and reduced bacterial motility. In addition to the structural data, these findings further highlight the importance of the terminal regions of FliD in mediating filament elongation through precise interactions with FliC, thereby stabilizing flagellins in the process of incorporation (Fig. 3d).

## Structure of the native HFJ

Using our short-flagellum cryo-EM dataset, we were able to determine the structure of the intact HFJ, anchored to both the hook and the filament, to 2.9 Å resolution (Extended Data Fig. 2). In this structure, both FlgK and FlgL form 11-mer layers that separate the hook protein FlgE from the filament protein FliC (Fig. 1d,e and Extended Data Fig. 6a). These components—FlgE, FlgK, FlgL and FliC—are aligned to form a continuous protofilament along the 11-start symmetry axis (Fig. 5a). The D0 domains of all four proteins face the lumen, creating a continuous channel with a consistent diameter of ~20 Å (Fig. 1e and Extended Data Fig. 6a). Helices in the D1 domain of all proteins, except FlgE, are entirely buried, whereas the D1 domain of FlgE; the D2 domains of FlgK, FlgL and FliC; and the D3 domain of FliC are exposed to the extracellular environment (Fig. 5a). The HFJ proteins follow a heterodimer pseudo-symmetry along the 11-start axis (Extended Data Fig. 6b).

FlgK and FlgL adopt three different modes to interact with adjacent proteins, corresponding to their position (Extended Data Fig. 6b–d). Interaction interface areas between each junction protein and their six adjacent subunits across three interaction modes were calculated using PISA[27,28] (Extended Data Table 2). As previously observed

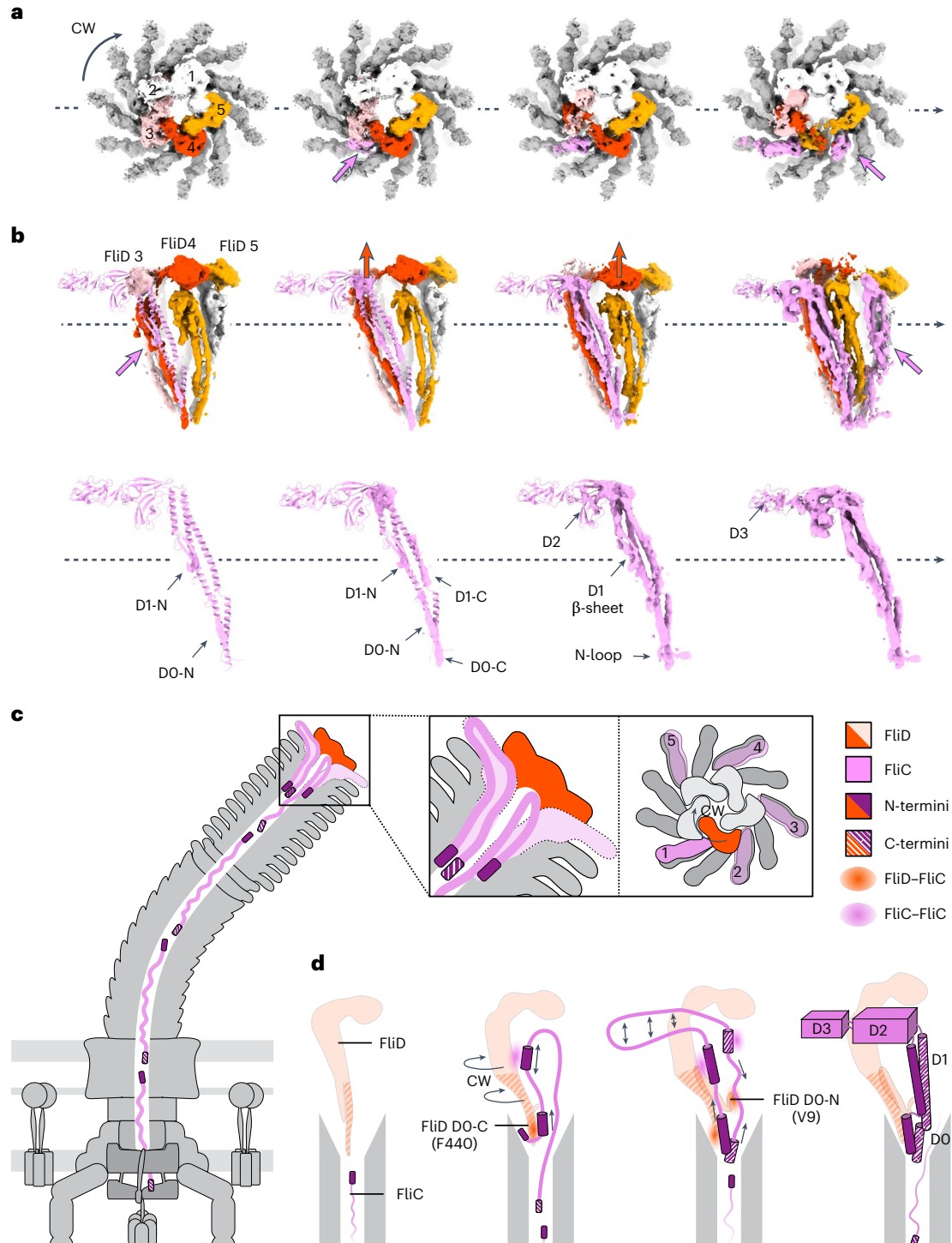

**Fig. 3 | Conformational changes to the cap and flagellin subunits upon filament assembly. a**, Top view of the CW rotation of the FliD cap (black arrow) and CCW incorporation of FliC (pink arrows). FliD subunits that undergo conformational changes are coloured salmon, red and orange. FliD subunits are labelled in the same order, as shown in Fig. 2c. **b**, Side view of the conformational changes of the FliD cap (red arrows) and FliC incorporation (pink arrows), with a focused view of the density appearing at each state of FliC incorporation. **c,d**, Mechanistic model of flagellin incorporation mediated by the FliD cap. **c**, Flagellin incorporation at the distal end of the bacterial flagellar filament facilitated by the FliD cap. Enlarged view on the left shows a simplified cross-section of the flagellar tip, while the right side provides a top-down view. Flagellin subunits are secreted in an at least partially unfolded state, with the N-terminus leading. The incorporation of a new flagellin subunit begins before the complete maturation of the previously inserted one. The FliD cap rotates in a CW direction,

while flagellin incorporation proceeds CCW. The FliD subunit highlighted in red facilitates the insertion of the flagellin subunit labelled '1', and the prospective sites for subsequent incorporation are numbered 2–5. **d**, Detailed view of the interactions involved in flagellin maturation and steps of flagellin folding. Interactions are visualized as coloured patches. Upon entering the FliD cap, the FliC N-terminus is captured by the D0-C domain of FliD. This interaction is followed by a 180° turn in the polypeptide chain, initiating the folding of the D0-N and D1-N domains of FliC, with stabilization provided by neighbouring FliC molecules. Simultaneously, the D0, D1 and D2 domains of FliD rise, accompanied by a CW shift of the D0–D1 domains. The FliD N-terminal loop then rotates 90° upwards, stabilizing FliC D0-C. Subsequent folding of FliC D1-C and D0-C occurs from both termini, with adjacent FliC subunits facilitating this process. Finally, the D2 and D3 domains are extruded from the cap cavity, completing the folding.

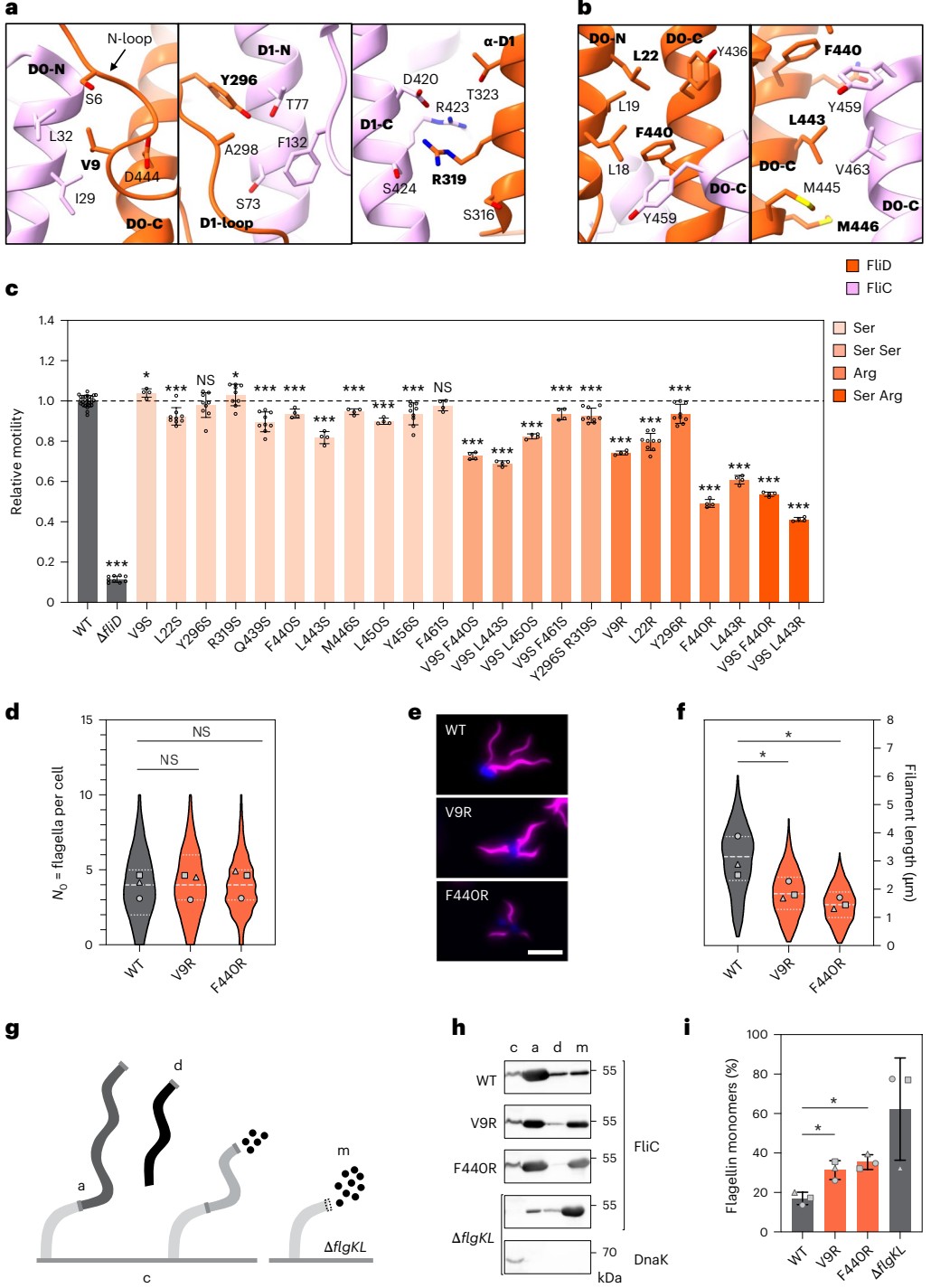

**Fig. 4 | FliD terminal regions mediate flagellin insertion at the flagellar distal end. a**, Interactions between FliD and FliC at position +5. Residues chosen for mutagenesis and related domains of FliD are highlighted (bold). **b**, Interactions between FliD and FliC at position +6. Residues chosen for mutagenesis and related domains of FliD are highlighted (bold). **c**, Relative motility of *S. enterica* FliD mutants analysed using soft-agar motility plates, 4–5 h of incubation at 37 °C. Motility halos were measured using Fiji, normalized to WT. The bar graphs show the mean ± s.d. of ≥3 biological replicates with individual data points. Ser, single substitution to serine; Ser Ser, double substitution to serine; Arg, single substitution to arginine; Ser Arg, double substitution to serine and arginine. **d**, Quantification of flagella per cell in WT and FliD mutants (V9R and F440R). Flagella per cell were counted for at least 150 cells per strain for 3 biological replicates ($n ≥ 465$). The violin plots show data distribution with median (dashed line) and quartiles (dotted lines); data points represent replicate means. **e**, Representative fluorescence microscopy images of WT and FliD mutants

(V9R and F440R). Filaments (FliC T237C) were labelled with Dylight555 Maleimide (magenta) and DNA counterstained with DAPI after pulsed *flhDC* induction. Scale bar, 3 μm. **f**, Quantification of filament length in WT and FliD mutants. Filament length was determined for at least 150 individual filaments per strain for 3 biological replicates ($n ≥ 605$). Data representation as in **d**. $P = 0.046$ (V9R); $P = 0.038$ (F440R). **g**, Schematic overview of experimental set-up to determine flagellin leakage. **h**, Representative immunodetection of cytoplasmic, cell-attached, detached and monomeric FliC in WT, FliD mutants (V9R and F440R) and $\Delta flgKL$. DnaK immunodetection of $\Delta flgKL$ shown as lysis control. c, cytoplasmic flagellin; a, attached flagellin; d, detached flagellin; m, monomeric flagellin. **i**, Proportion of secreted flagellin as a percentage of total flagellin amount of WT, FliD mutants (V9R and F440R) and $\Delta flgKL$. The bar graphs show the mean ± s.d. of three biological replicates. $P = 0.018$ (V9R); $P = 0.010$ (F440R). Statistical annotations were calculated with a two-tailed Student's *t*-test, on the means of biological replicates (*$P < 0.05$; **$P < 0.01$; ***$P < 0.001$; NS, non-significant).

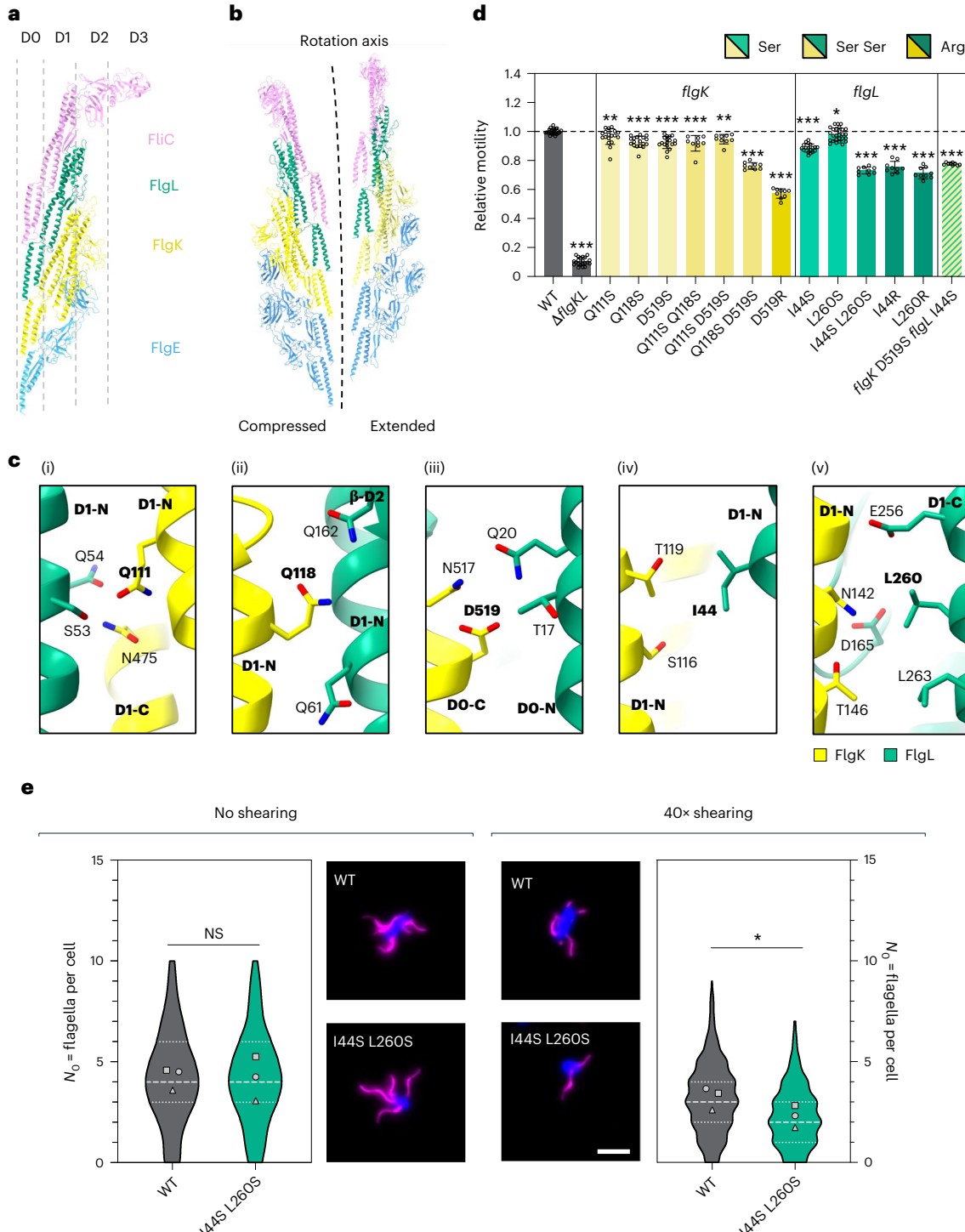

**Fig. 5 | Mutations in the HFJ interface impair bacterial motility and filament stability. a**, Side view and domain organization of the protofilament of FlgE–FlgK–FlgL–FliC (PDB: 9GO6). **b**, Cross-sectional overview of the compressed protofilament and extended protofilament of HFJ in complex with the hook and the filament with the labelled rotation axis. **c**, Molecular details of interfaces between FlgK and FlgL. Residues Q111 (i), Q118 (ii) and D519 (iii) for FlgK and I44 (iv) and L260 (v) for FlgL are found to interact with their neighbouring residues in PISA. Mutated residues and the domains they are located in are highlighted in bold black. **d**, Relative motility of *S. enterica* FlgK and FlgL point mutants analysed using soft-agar motility plates, quantified after 4–5 h of incubation at 37 °C. Diameters of the motility halos were measured using Fiji and normalized to the WT. The bar graphs represent the mean of at least three biological replicates with standard deviation error bars. Replicates are shown as individual data points. Ser, single substitution to serine; Ser Ser, double substitution to

serine; Arg, single substitution to arginine. **e**, Quantification of the number of flagella per cell in the WT and FlgL I44S L260S mutant including representative fluorescence microscopy images (middle) without shearing (left) and after 40× shearing (right) using the pulsed *flhDC* induction set-up. Filaments (FliC T237C) were labelled with Dylight555 Maleimide (magenta) and DNA counterstained with DAPI. Scale bar, 2 μm. The number of flagella per cell ($N_0$) was determined for at least 150 cells per strain and condition for 3 biological replicates ($n \geq 634$). The violin plots represent the distribution of the data of all replicates including the median (dashed line) and quartiles (dotted lines). Data points represent the means of each biological replicate. $P = 0.937$ (no shearing) and $P = 0.049$ (40× shearing). Statistical annotations were calculated with a two-tailed Student's *t*-test on the means of each biological replicate (*$P < 0.05$; **$P < 0.01$; ***$P < 0.001$; NS, non-significant).

for the hook[3,30], the protofilaments in the junction can be structurally classified into two types: extended and compressed (Fig. 5b). To better understand the mechanical role of the HFJ (FlgKL) in torque transmission, we applied 3DVA to visualize the flexibility transitions from the hook to the filament (Supplementary Video 5 and Extended Data Fig. 7g–j). The molecular structure of FlgE–FlgK–FlgL–FliC protofilaments, as well as pairs of each component in both the extended and compressed states, were compared (Extended Data Fig. 7a–d and Supplementary Video 6). Substantial structural shifts occur in FlgK, FlgL and FlgE subunits when transitioning from the extended to the compressed state, while the shift in flagellin subunit FliC remains minimal. During this transition, the D0 domain and the uppermost part of the D2 domain of FlgK bend towards the lumen (Extended Data Fig. 7a). Similarly, the D0 domain of FlgL rises and bends slightly towards the lumen (Extended Data Fig. 7b). The D0 and D2 domains of FlgE become curved (Extended Data Fig. 7c), consistent with previous EM structures showing that FlgE gains flexibility through flexible hinges connecting D0 and D2 to D1 (ref. 3). Notably, FliC shows no substantial shift (Extended Data Fig. 7d), suggesting that the HFJ prevents mechanical force transmissions to the filament. In our molecular model, not all helices in the D1 domain of FlgK are straight and continuous. This discontinuity may provide FlgK with the elasticity needed to absorb mechanical stress. We propose that the HFJ functions as a mechanical buffer, leveraging the elasticity of FlgK to prevent mechanical stress from propagating to the flagellin filament.

To investigate the assembly of the HFJ, we modelled various combinations of its components (Extended Data Fig. 7e,f). We questioned whether the transition between the hook and the filament remains stable when FliC is assembled onto the FlgE hook without the HFJ or when only a single layer of FlgL is present. When FlgK is substituted with FlgL or FliC, a large gap forms at the top of FlgE (Extended Data Fig. 7e). This discontinuity may destabilize the connection between the hook and filament, suggesting that an intermediate element, such as FlgK, is necessary. Next, we investigated whether FliC could assemble directly onto the FlgK layer and whether FlgK or FlgL could form multilayered structures. Structural modelling revealed notable clashes when FlgL was substituted with FlgK in the FlgKL complex, indicating that multilayered FlgK assemblies are inherently unstable (Extended Data Fig. 7f). Unexpectedly, no obvious clashes were observed in the FliC–FlgK or double-layer FlgL configurations, indicating structural feasibility, although this was not supported by our experimental data.

## HFJ mutations impair motility and filament stability

On the basis of interaction analysis using PISA[27,28], residues Q111, Q118 and D519 for FlgK, and I44 and L260 for FlgL, were determined to be buried at the interface between FlgK and FlgL (Fig. 5c). To test the role of these residues in HFJ function, we constructed mutants of these residues and assessed their ability to swim through semi-solid agar (Fig. 5d).

Substitution of these FlgK residues to serine has only a mild effect on motility, with 4–8% reduction compared with the WT (Q111S, Q118S and D519S). When double mutations are introduced (Q111S Q118S, Q111S D519S, and Q118S D519S), only Q118S D519S shows a stronger reduction compared with the respective single serine mutants. The D519R substitution led to the most significant reduction of motility to 57% compared with the WT. As FlgK interacts with its chaperone FlgN via its C-terminal region[31], we confirmed that the observed motility defects were not caused by impaired secretion (Extended Data Fig. 8a). Both FlgK D519S and FlgK D519R mutants were secreted at WT levels.

For FlgL, the single serine mutants I44S and L260S show mild reductions of motility, to 88% and 97% of WT motility, respectively (Fig. 5d). However, in the FlgL double mutant (I44S L260S), motility is reduced to 73% compared with the WT. Similarly, substituting the selected residues of FlgL to arginine (I44R, L260R) resulted in

greater reductions to 75% and 71% WT motility, respectively. Combining the serine mutations FlgK D519S and FlgL I44S reduced the motility to 77%. In conclusion, our results indicate that, while the interface appears relatively robust towards mutations, the integrity of the FlgKL interface is crucial for maintaining motility in *S. enterica*. Mutations to larger, charged residues or double mutations of highly conserved residues within this interface led to significant reductions in motility, supporting our hypothesis that these interactions are vital for the stability of the HFJ.

To assess whether FlgKL mutations destabilize HFJs, we performed a filament shearing assay (Fig. 5e and Extended Data Fig. 8b). Fluorescently labelled filaments were sheared by passing the cells through a narrow needle, and the resulting flagellation patterns were analysed by fluorescence microscopy. Without shearing, the FlgL I44S L260S mutant shows the same median number of filaments per cell as the WT (four flagella per cell). However, after 40× shearing, the WT shows a median of three flagella per cell, whereas the FlgL double mutant displays a significantly decreased median of only two flagella per cell. These results support our hypothesis that FlgKL interface mutations lead to a less stable HFJ, as evidenced by the increased susceptibility to filament breakage under mechanical stress. This instability probably contributes to the observed motility defects, underscoring the importance of FlgKL interactions for the structural integrity and proper function of the flagellar apparatus.

## Structure of the cap complex bound to the HFJ

The cap complex was shown to assemble on the HFJ[32,33], which acts as a priming step to initiate filament assembly. However, the molecular details of the interaction between the HFJ and the cap at the initial stage of flagellin incorporation remain unclear. To investigate this, we aimed to determine the structures of the cap complex bound to the HFJ in the absence of the filament. Specifically, we exploited a *C. jejuni* Δ*flhG* minicell strain with additional deletions of the flagellin genes *flaA* and *flaB*[34]. In these minicells, flagella consist of only the basal body, the hook, the HFJ and the FliD cap (Fig. 6a). We manually picked the hook tips to generate the template for the following automatic picking, leading to 79,106 particles. Particles were then subjected to several rounds of two dimensional (2D) classification (Fig. 6b). This allowed us to produce a map of the FliD cap of *C. jejuni* assembled on the HFJ to an overall resolution of 6.5 Å with 15,077 particles (Fig. 6c and Supplementary Table 1), and the atomic model of the cap–HFJ complex was built (Fig. 6d). In this structure, the cap complex also consists of a FliD pentamer with an overall width of ~130 Å and a length of ~180 Å—dimensions that are comparable to the cap structure bound to the filament (Figs. 1d,e and 6e). Like our previous observations, the HFJ of *C. jejuni* shows a two-layer structure, with the proximal layer comprising 11 FlgK subunits and the distal layer comprising 11 FlgL subunits (Fig. 6e,f). Notably, the *C. jejuni* FliD orthologue includes an additional domain, D4, not present in *S. enterica* (Fig. 6g). D4 domains are structurally distant from the cap cavity, suggesting that they are not involved in flagellin incorporation. Most FliD monomers adopt an angle of 89–95° between the D0–D1 and D2–D3 axes, which is larger compared with the angles of 81–84° measured from the filament-bound FliD (Figs. 2c and 6g). FliD 3 in *C. jejuni* shows an angle of 82°, compared with 56° in *S. enterica*. Importantly, the observed 56° D1–D2 angle of FliD 3 in *S. enterica* reflects an ongoing incorporation cycle, whereas the 82° angle in *C. jejuni* represents a pre-incorporation state, in which the cap is positioned on the HFJ but not yet actively engaged in flagellin polymerization. When the two cap complex structures are superimposed, it can be seen that their structures are highly similar. However, the D2–D3 plane of the *C. jejuni* cap is flat compared with the tilted plane of the *S. enterica* cap (Fig. 6h(i–iii)) while the D0 and D1 helices are well aligned (Fig. 6h(iv)). On the basis of our structure of the *C. jejuni* FliD pentamer, the D4 domain of the next FliD in the CCW direction is located between the D3 domain and an

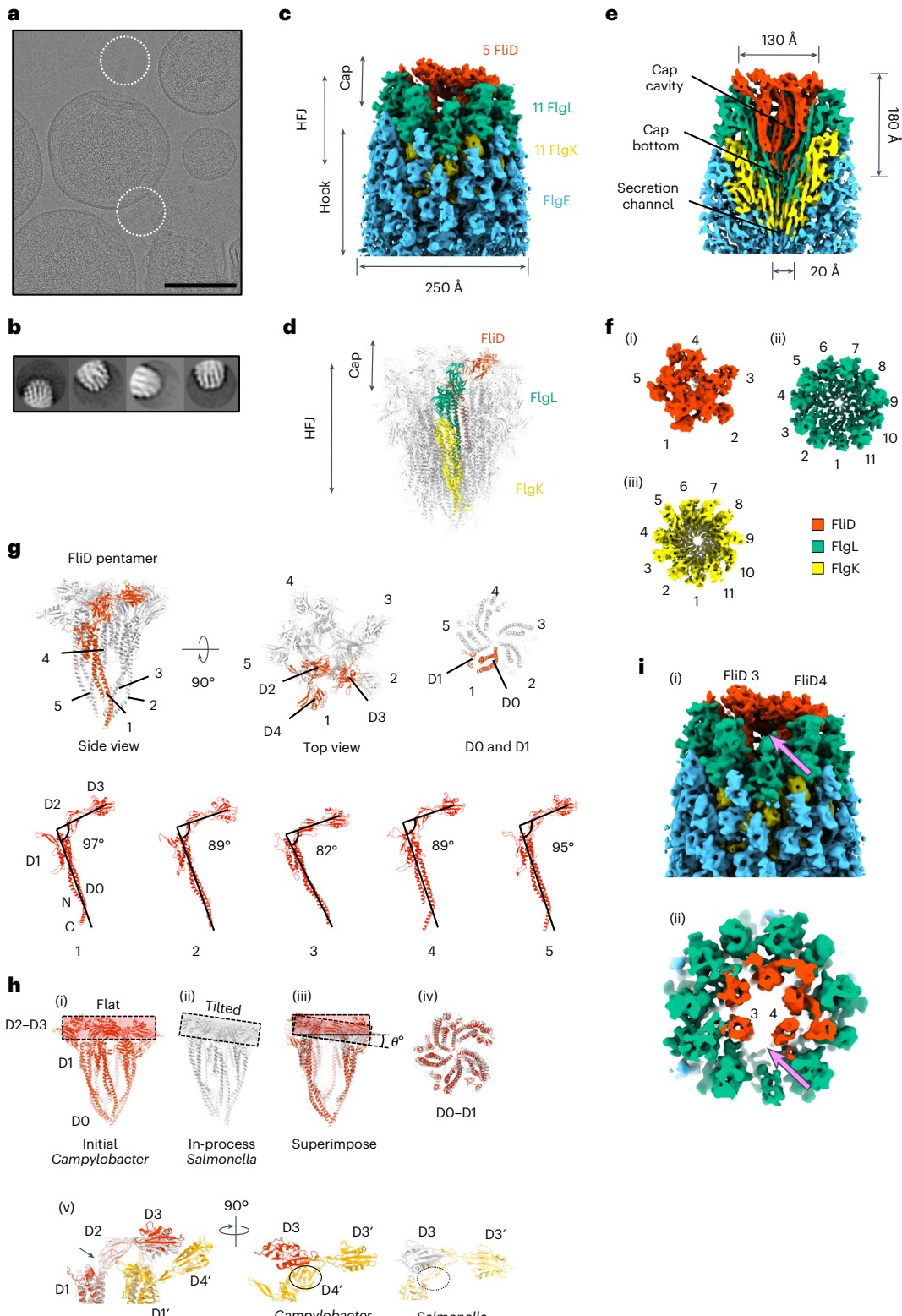

**Fig. 6 | Structure of the cap complex bound to the HFJ. a**, *C. jejuni* minicells in cryo-EM showing flagellar tips (EMPIR-11580). Scale bar, 200 nm. **b**, Representative 2D classes of *C. jejuni* flagellar hook tips. **c**, Density map of the *C. jejuni* tip complex that contains the hook, the HFJ and the cap. Components are labelled and their stoichiometry is indicated. **d**, Atomic model of the *C. jejuni* cap–HFJ complex (PDB: 9GSX). The protofilament of FlgK–FlgL and the adjacent FliD subunit is highlighted. **e**, Cross-section view of the density map of **c** with labels for the dimension of the FliD cap. **f**, Map segment and corresponding stoichiometry of each section in the hook–junction–cap complex. (i) FliD pentamer. (ii) FlgL layer. (iii) FlgK layer. **g**, Overall atomic structure of the FliD

cap of *C. jejuni* and structural polymorphism among FliD. **h**, Comparison between the *C. jejuni* cap structure and the *S. enterica* cap structure. (i) Structure of the *C. jejuni* cap with the flat D2–D3 plane highlighted. (ii) Structure of the *S. enterica* cap with the tilted D2–D3 plane highlight. (iii) Structural superimposition of the *C. jejuni* cap and *S. enterica* cap. The angle difference in their D2–D3 plane is indicated. (iv) Superimposition on their D0–D1 domains. (v) Superimposition of the D2–D4 domains of FliD 3 (red) and FliD 4 (orange). The D1–D2 hinge is indicated by the arrow. **i**, Side view (i) and cross-section (ii) of the density map of the *C. jejuni* tip. A gap is found between FliD 3 and FliD 4 that is primed for the first FliC to be incorporated, indicated by pink arrows.

FlgL subunit (Fig. 6h(v)). This could explain why the angles of *C. jejuni* FliDs are generally larger than those of *S. enterica* FliDs. For FliD 3, the D4 domain of FliD 4 and the bulky D2 domain of FlgL may prevent the D3 domain from occupying the gap between two FlgL subunits (Fig. 6i). By contrast, *S. enterica* lacks a D4 domain in FliD and a bulky D2 domain in FlgL, allowing the D3 domain to move further down to occupy the gap.

In contrast to the filament-bound cap structure, one of the FliD monomers (FliD 3) is positioned downwards, and its D2–D3 domains are flipped, probably corresponding to a major conformational change from initiation to elongation (Supplementary Video 7). Additional conformational changes occur in other FliD molecules, including in the D2 and D3 domains that are positioned more upwards, resulting in a flat D2–D3 plane. Nonetheless, we cannot exclude the possibility that some of these structural differences may correspond to species-specific distinct conformations.

Interestingly, there is a gap between FliD 3 and FliD 4 in the cap bound to the HFJ (Fig. 6i), corresponding to the position of FliC incorporation in the structure of the cap bound to the filament. We propose that this gap is the first FliC incorporation site, and therefore, the first FliC corresponds to the FliC at +11 on top of FlgL 1 (Fig. 6i and Extended Data Fig. 6b).

## Discussion

Here we report complete structures of the extracellular bacterial flagellum of *S. enterica* and *C. jejuni*, providing critical insights into filament growth initiation and elongation mediated by the filament cap. Our structural and mutational analyses also underscore the HFJ as an essential linker between the hook and the filament, consistent with a recent study[35].

Consistent with earlier studies, our findings confirm that the filament cap rotates CW during filament growth (Fig. 3a), undergoing asymmetric structural rearrangements to facilitate flagellin incorporation[17,20]. Concomitantly, the incorporation of flagellin subunits occurs in the CCW direction (Fig. 3a,c), in line with earlier low-resolution data[17]. We propose a refined model in which the N-terminus of FliC initially interacts with the FliD D0-C domain and already polymerized flagellins, anchoring and guiding subsequent folding of flagellin domains (Fig. 3d). This guided incorporation ensures efficient assembly in vivo, highlighting the critical chaperone-like function of the filament cap. The efficient incorporation of flagellins mediated by both N- and C-terminal regions of the filament cap is essential, given the high metabolic cost associated with the entire process of flagellum assembly. Notably, comparable structural strategies are used by different cap proteins (rod cap FlgJ and hook cap FlgD) despite sequence divergence, suggesting similar mechanisms facilitating rod, hook and filament assembly (Supplementary Discussion)[25,36–38].

Our results also confirm the mechanical importance of the HFJ, showing that mutations disrupting FlgKL interfaces significantly impair motility and the structural integrity of the filament, in line with earlier observations in *E. coli*[39]. Notably, we show that the HFJ serves a dual function, acting both as a connecting platform and as a mechanical buffer, preventing mechanical stress originating from hook flexibility from destabilizing the filament. This role is particularly crucial given the distinct mechanical properties of the hook, which is flexible in bending yet rigid in twisting and allows rotational movement and the transmission of torque[3,30,40]. By contrast, the rigid filament undergoes rapid structural changes, alternating between left-handed and right-handed supercoils in response to the changing direction of motor rotation (Supplementary Discussion)[41,42]. Variations observed between bacterial species, such as differences in hook rigidity between *C. jejuni* and *S. enterica*, suggest evolutionary adaptation of HFJ and hook proteins to specific mechanical and functional requirements[35,43].

The HFJ is thought to remove the hook cap FlgD, assemble cap-less and then facilitate the formation of the filament cap for filament elongation; however, the exact assembly mechanism remains elusive[44]. Our structural data suggest a sequential CW assembly of FlgK followed by FlgL, required to stabilize the emerging filament cap composed of FliD. The elasticity introduced by the FlgK layer probably explains why direct assembly of flagellin onto FlgK is structurally unfavourable, necessitating FlgL as a stabilizing intermediate.

The mechanism proposed here significantly advances the understanding of bacterial flagellar filament assembly and clarifies key molecular events underpinning flagellar filament growth, highlighting structural adaptations and mechanistic principles broadly applicable across diverse bacterial species.

## Methods

### *S. enterica* strains and cultivation

All *S. enterica* strains used in this study are listed in Supplementary Table 2 and were derived from *S. enterica* subsp. *enterica* serovar Typhimurium LT2. Bacteria were grown under constant shaking at 180 rpm in lysogeny broth (LB) at 37 °C, unless stated otherwise, and supplemented with 100 ng ml$^{-1}$ anhydrotetracycline (AnTc) when required. Bacterial growth was determined by measuring the optical density at 600 nm using a spectrophotometer (Amersham Bioscience). For transductional crosses, the general transducing *Salmonella* phage P22 HT105/1 int-201 was used[45]. Point mutations and gene deletions were introduced into the *S. enterica* genome to maintain native expression conditions using lambda-red homologous recombination[46]. The oligonucleotides used for strain construction are listed in Supplementary Table 3.

### Purification of short flagella

The purification of short flagella from *S. enterica* strain EM16009 (Supplementary Table 2) was adapted from a previously published protocol[24–26]. Briefly, an overnight culture was inoculated with a single colony in LB medium. The next day, 500 ml of LB medium was inoculated 1:100 with the overnight culture and grown for 3 h. To induce flagellin production, AnTc was added, and the cells were further inoculated for 30 min. Cells were collected at 4,000 × *g* at 4 °C for 15 min. The cell pellet was carefully resuspended in 20 ml of ice-cold sucrose solution (0.5 M sucrose, 0.15 M Trizma base, unaltered pH) on ice. Lysozyme and EDTA (pH 4.7) were added slowly to final concentrations of 0.1 mg ml$^{-1}$ and 2 mM, respectively, while stirring the cell suspension on ice. After 5 min of stirring on ice, the suspension was transferred to room temperature and slowly stirred for 1 h to allow spheroplast formation. For cell lysis, *n*-dodecyl β-maltoside (DDM) was added at a final concentration of 0.5%. After the suspension was rapidly stirred for 10 min until it became translucent, it was slowly stirred further for 30–45 min on ice. To degrade DNA, 2 mg DNAseI and MgSO$_4$ were added to a final concentration of 5 mM while stirring on ice. After 5 min, EDTA (pH 4.7) was added at a final concentration of 5 mM. To pellet cell debris and unlysed cells, the suspension was centrifuged at 15,000 × *g* at 4 °C for 10 min. The supernatant was collected and centrifuged at 100,000 × *g* for 1 h at 4 °C to pellet the flagella. The flagella were washed carefully with 30 ml buffer A (0.1 M KCl, 0.3 M sucrose, 0.05% DDM) and collected again at 100,000 × *g*. The flagella were resuspended carefully in 100 µl buffer B (10 mM Tris (pH 8), 5 mM EDTA (pH 8), 0.01% DDM) and incubated overnight on a rolling platform at 4 °C. The next day, samples were used for cryo-EM grid preparation.

### Single-particle cryo-EM sample preparation and data acquisition of *S. enterica* cap and HFJ

Resuspension of short flagella (3 µl) was applied to glow-discharged holey carbon grids (Quantifoil R2/2, 300 mesh). Samples were incubated for 30 s at 4 °C and 88% humidity before being blotted by Leica EM GP1 and then rapidly plunged into liquid ethane. Grids that were blotted for 3, 4, 5 and 6 s were screened on a 200 kV Glacios microscope (Thermo Fisher). The grids with good ice thickness were deposited to

a 300 kV Krios G3i microscope with a Gatan K3 direct electron detector (Thermo Fisher). The dataset was collected using a physical pixel size of 1.078 Å at a magnification of ×81,000. Finally, 24,729 videos were collected at a dose rate of 16.9 e– pix$^{-1}$ s$^{-1}$ and exposure of 3.2 s, corresponding to a total dose of 43 e– Å$^{-2}$. All videos were collected over 40 frames with a defocus range of −0.9 µm to −2.7 µm.

## Single-particle cryo-EM image processing and EM map reconstruction of the *S. enterica* cap and HFJ

For all videos, motion correction and contrast transfer function estimation were processed in CryoSPARC v4.4 (ref. 47) using patch motion correction and patch CTF estimation, respectively. A total of 24,066 videos were used.

For the FliD cap, 1,896 particles were manually picked and subjected to 2D classification to generate templates, followed by template-based automatic picking. A total of 527,490 particles were recognized and extracted with a box size of 500 × 500 pixels. Multiple rounds of 2D classification were performed, leading to a subset of 63,208 particles. After a homogeneous refinement, local CTF refinements and a round of non-uniform refinement, a 3.3-Å-resolution map of the FliD cap complex was obtained. The 3DVA (ref. 29) was performed to analyse discrete heterogeneity and resolve continuous flexibility. A total of 42,342 particles were used for the analysis, with 3 variability components computed. The analysis was conducted with a spatial filter of 8 Å and a mask encompassing the entire FliD pentamer along with its adjacent FliC subunits. No high-pass filter was applied during the process.

To further resolve the density of the D2–D3 of the most flexible FliD, 3D classification with 15 classes was performed attempting to discriminate different transition states of the FliD cap during elongation, which resulted in 8 different states of the FliD cap. A subset of 15,225 particles was selected from the best class followed by non-uniform refinement and local refinement with a mask where D2–D3 domains of flagellin are removed, leading to a 3.7-Å-resolution map with all FliD subunits well resolved.

For the HFJ, filament tracing was performed initially, with a filament diameter of 220 Å and a separation distance of 0.25-fold diameters. A subset of flagellin filament obtained from the initial 2D classification of the FliD cap was used as the template for the filament tracer. A total of 3,349,411 particles were extracted with a box size of 500 × 500 pixels and subjected to 2D classification. Several rounds of 2D classification were performed to remove particles that did not contain the HFJ, and a subset of 125,307 particles was selected. After homogeneous refinement, a round of local CTF refinement and a round of non-uniform refinement, a map with an average resolution of 2.5 Å was obtained. However, in this initial map, the density of the HFJ appeared to overlap with features of the FlgE hook and FliC filament, probably owing to particle misalignment. To resolve this, we performed 3D classification into 15 classes to distinguish junctions at different relative positions within the density map. This analysis resulted in two major junction classes containing 65,561 and 58,875 particles. Subsequent non-uniform refinement of each class yielded improved density maps at 3.0-Å and 3.2-Å resolution, respectively, resolving the issue of mixed particle populations. A local refinement with a tight mask was performed based on the 3.0-Å junction map, which improved the average resolution of the junction to 2.9 Å and the occupancy of FlgK and FlgE. For the visualization of the transition from the extended to the compressed state of the HFJ, we performed 3DVA on a total of 65,561 HFJ particles with five variability components, a spatial filter of 8 Å and a mask covering the entire HFJ map. No high-pass filter was applied.

## Atomic model building and refinement of the *S. enterica* cap and HFJ

For the FliD–FliC model, the monomer structure of the full-length FliD and FliC was generated using AlphaFold3 (ref. 48). First, the D0–D1 domain and D2–D3 domain of each FliD were isolated and manually

fitted into the reconstructed maps as rigid bodies, and the FliD pentamer was built. Flexible fitting was then performed on the FliD pentamer using ISOLDE[49] with secondary structure restraints gained from the AlphaFold model in UCSF ChimeraX[50]. After the flexible fitting was completed, D0–D1 and D2–D3 of each FliD were connected automatically in PyMOL. FliC flagellins were then replicated and manually fitted into the map as rigid bodies. ISOLDE was used again to flexibly fit the whole complex of 5 FliD and 17 FliC into the reconstructed map. The FliD–FliC complex model was refined with real-space refinement in PHENIX[51] with secondary structure, rotamer and Ramachandran restraints but without non-crystallographic symmetry restraints. Coot was then used to correct rotamer outliers, side-chain clashes and unattributed density[52]. The final model was validated using a comprehensive validation program with default settings in PHENIX. For the hook–filament junction model, monomer structures of the full-length FlgE, FlgK, FlgL and FliC were generated using AlphaFold3 (ref. 48). All subunits were manually fitted into the reconstructed maps as rigid bodies at first, including 13 FlgE, 11 FlgK, 11 FlgL and 14 FliC. Then, ISOLDE was used for flexible fitting with secondary structure restraints for the entire complex, followed by the same refinement and validation procedures and settings we used to refine the FliD–FliC model using PHENIX and Coot.

The *Q*-scores of the *Salmonella* filament cap and HFJ models were approximately 0.4. Local resolution estimations (Extended Data Fig. 2) indicate that the resolution of D2–D3 is lower than that of D0–D1, which accounts for the somewhat lower *Q*-scores, particularly in these regions.

## Swimming motility

Swimming motility was studied using tryptone broth-based soft agar swim plates containing 0.3% Bacto agar. Motility plates were inoculated with 2 µl of overnight culture and incubated at 37 °C for 4–5 h. Images were acquired by scanning the plates, and the diameters of the swimming halos were measured using Fiji[53]. The swimming diameters of the mutant strains were normalized to those of the WT.

## Pulsed *flhDC* induction set-up

To synchronize flagellar biosynthesis and, when required, prevent the negative feedback loop of FliT on FlhDC, we used strains with an AnTc inducible promoter for the master regulator FlhDC (P$_{tetA}$-*flhDC*). Flagellum formation is regulated by multiple feedback loops, involving the chaperone FliT[54]. In the cytosol, FliT binds to FliD, targeting it to the export gate. Upon hook basal body completion and secretion of FliD, FliT is released and binds to the FlhC subunit of the flagellar master regulator, preventing expression of flagellar class 2 and subsequently class 3 genes[54]. Previous studies have shown that the C-terminal region of FliD binds FliT[55]. To minimize putative effects of the FliT feedback loop in our C-terminal FliD mutants and study flagellin polymerization, we synchronized flagellum assembly by expressing *flhDC* from an inducible promoter (P$_{tetA}$-*flhDC*)[56]. We reasoned that cellular levels of free FliT would be low during initial rounds of flagellum assembly, as unbound FliT accumulates over time.

Briefly, we induced *flhDC* expression with a 30-min pulse of AnTc, followed by removal of the inducer and a 60-min incubation step. This pulsed *flhDC* induction set-up was used for fluorescence microscopy, flagellin leakage, filament shearing and secretion assays (Extended Data Fig. 5e). Overnight cultures were grown in the absence of AnTc and were diluted 1:10 in a total volume of 10 ml at 30 °C for 1.5 h before inducing flagellar biosynthesis with the addition of AnTc. Cells were grown for 30 min in the presence of AnTc at 30 °C. Cells were centrifuged for 5 min at 2,500 × *g* to remove the inducer, resuspended in 10 ml fresh LB and incubated further for 60 min at 30 °C before sample collection.

## Fluorescence microscopy

All strains investigated by fluorescence microscopy, except for EM16009 that was used for purification of short flagella, were locked

in the expression of *fliC* (Δ*hin*-5717::FRT) and contained a cysteine mutation in FliC (*fliC*6500 T237C), which allowed labelling of the flagellin subunit with a fluorophore-coupled maleimide dye[57]. After the cells were cultured using the pulsed *flhDC* induction set-up, 500 µl of cell suspension was collected. Cells were centrifuged for 5 min at 2,500 × *g* and resuspended in 500 µl of 1 × phosphate-buffered saline (PBS). Fluorophore-coupled maleimide dye (Alexa Fluor 488, Invitrogen) was added at a final concentration of 10 µM, and cells were further incubated for 30 min at 30 °C. Following the removal of unbound dye by slow-speed centrifugation for 5 min at 2,500 × *g*, cells were resuspended in 500 µl 1 × PBS and applied to a homemade flow cell. Flow cell preparation was performed as previously described[58]. Briefly, coverslips were incubated with 0.1% poly-L-lysine for 10 min, air-dried for 10 min and subsequently fixed to an objective slide via two layers of preheated parafilm to create a chamber. The side of the coverslip incubated with poly-L-lysine faced the objective slide. Cells were allowed to adhere to the coverslip for 3 min at room temperature in the dark and were subsequently fixed with 2% (v/v) formaldehyde and 0.2% (v/v) glutaraldehyde for 10 min. Cells were washed twice with 1 × PBS and finally mounted with Fluoroshield + DAPI solution (Sigma-Aldrich). Images were acquired by an inverted epifluorescence microscope (Zeiss AxioObserver.Z1) at ×100 magnification with a Prime BSI Scientific CMOS (sCMOS) camera, a Plan Apo 100×/1.4 Oil Ph3 objective and a LED Colibri 7 light source (Zeiss). Images were taken using the Zen 3.8 Pro software with *Z*-stack every 0.5 µm with a range of 3 µm (7 slices). Images were analysed using Fiji[53] and the NeuronJ plugin[59].

To test the optimal AnTc induction time for flagellin production and purification of short flagella (Extended Data Fig. 1d), an overnight culture of EM16009 was diluted 1:100 and grown for 3 h before the induction of FliC production with AnTc for 30, 45, 60 or 120 min. At each time point, 500-µl samples were collected, loaded onto a flow cell and fixed as described above. Cells were washed twice with 1 × PBS and blocked with 10% bovine serum albumin (BSA) for 10 min. Primary α-FliC antibodies (BD Difco *Salmonella* H Antiserum i, catalogue number 11712894, 1:1,000 in 2% BSA) were added for 1 h. Cells were washed twice with 1 × PBS and subsequently blocked with 10% BSA for 10 min. Secondary antibody (anti-rabbit-Alexa488, Invitrogen, catalogue number A-11034, 1:1,000 in 1 × PBS) was added for 30 min. Cells were washed twice with 1 × PBS and mounted with Fluoroshield + DAPI solution (Sigma-Aldrich). Fluorescence microscopy was carried out as described above.

## Single-cell tracking

Bacteria were diluted 1:100 from an ON culture and grown in LB medium at 37 °C with shaking for 1.5 h. When required, FliC expression was induced with AnTc. Following induction, samples were taken after 30 min and 120 min. Samples were diluted to an OD$_{600}$ of 0.1 in 1 × PBS supplemented with 0.2% glucose and 0.1% Tween20. Aliquots of the cell suspensions were loaded into an uncoated flow cell. The swimming behaviour was recorded using an inverted microscope (Zeiss AxioObserver.Z1) at ×20 magnification with a Prime BSI Scientific CMOS (sCMOS) camera and a Plan-Apochromat ×20/0.8 Ph2 objective. Videos were taken using the Zen 3.8 Pro software for a duration of 10 s at 33-ms intervals. Videos were batch processed and analysed using a custom Jupyter Notebook, Ilastik v1.4.0 (ref. [60]) and Fiji[53] equipped with the TrackMate plugin[61] as described previously[23]. Briefly, the pixel classifier in Ilastik was used to distinguish cells from the background. The resulting segmentation masks were exported for further analysis via a custom python script, in which bacterial trajectories were tracked, and single-cell velocities were computed using the Simple LAP tracker available in TrackMate. Tracks shorter than 1 s were excluded. The mean swimming speed was extracted for each track.

## Flagellin leakage assay

After the cells were cultured using the pulsed *flhDC* induction set-up, a 3-ml aliquot was taken, and cells were collected through centrifugation

at 13,000 × *g* for 5 min. The samples were further treated to separate monomeric FliC from cytoplasmic, cell-associated and cell-detached FliC (Fig. 4e and Extended Data Fig. 5e). Subsequently, 2 ml of the supernatant was transferred to a fresh tube, the remaining supernatant was removed and the cell pellet was resuspended in 1.5 ml 1 × PBS. The cell pellets were incubated at 65 °C for 5 min to depolymerize the flagellar filament, and the samples were subsequently centrifuged to obtain cell pellets and supernatants containing cytoplasmic flagellin molecules and depolymerized flagellin monomers, respectively. The supernatants were ultracentrifuged at 85,000 × *g* for 1 h at 4 °C, and the pellets containing flagellar filaments detached from the cell bodies and supernatants containing flagellin monomers leaked into the culture medium were collected separately. Proteins from different fractions were precipitated using 10% trichloroacetic acid (TCA) and separated by SDS-PAGE. FliC protein levels in the different samples were determined by immunoblotting using primary α-FliC (Difco, catalogue number 228241 *Salmonella* H Antiserum I, 1:5,000 in 1 × TBS-T) and secondary α-rabbit antibodies (Bio-Rad Immun-Star Goat Anti-Rabbit-HRP Conjugate, catalogue number 170-5046, 1:20,000 in 1 × TBS-T). Relative FliC protein levels were normalized to the housekeeping protein DnaK, which was detected using α-DnaK (Abcam, catalogue number ab69617, 1:10,000 in 1 × TBS-T) antibodies and secondary α-mouse antibodies conjugated to horseradish peroxidase (Bio-Rad Immun-Star Goat Anti-Mouse-HRP Conjugate, catalogue number 170-5047, 1:20,000 in 1 × TBS-T) antibodies, using the Image Lab software (Bio-Rad). The percentage of flagellin monomers secreted into the culture supernatant was calculated by dividing the amount of secreted flagellin monomers by the total flagellin amount, comprising secreted and cellular as well as detached and attached flagellin molecules.

## Filament shearing assay

All strains investigated in the shearing assay were locked in the expression of *fliC* (Δ*hin*-5717::FRT) and contained a cysteine mutation in FliC (*fliC*6500 T237C), which allowed labelling of the flagellin subunit with a fluorophore-coupled maleimide dye[57]. After the cells were cultured using the pulsed *flhDC* induction set-up, 1,000 µl of the cell suspension was collected. Cells were centrifuged for 5 min at 2,500 × *g* and resuspended in 500 µl of 1 × PBS. Fluorophore-coupled maleimide dye (Alexa Fluor 488, Invitrogen) was added at a final concentration of 10 µM, and cells were further incubated for 30 min at 30 °C. Following the removal of unbound dye by slow-speed centrifugation for 5 min at 2,500 × *g*, the cells were resuspended in 1,000 µl 1 × PBS. Samples were split into 2 × 500-µl aliquots. One sample was left untreated, while the flagella of the other sample were sheared by passing the cell suspension 40× back and forth through a 27-G, 0.4 × 12 mm BL/LB needle using a 1-ml syringe. For the first replicate (Extended Data Fig. 8b), the samples were split into 4 × 500-µl aliquots. Again, one sample was left untreated, while the flagella of the other samples were sheared by passing the cell suspension 20×, 40× or 60× back and forth through a 27-G, 0.4 × 12 mm BL/LB needle using a 1-ml syringe. To separate the cells from the sheared filaments, the samples were centrifuged at 2,500 × *g* and the supernatant containing most of the sheared filaments was discarded. Cells were resuspended in 500 µl of 1 × PBS and applied to a homemade flow cell as described above. Images were acquired by an inverted epifluorescence microscope (Zeiss AxioObserver.Z1) at ×100 magnification with a Prime BSI Scientific CMOS (sCMOS) camera, a Pln Apo ×100/1.4 Oil Ph3 objective and a LED Colibri 7 light source (Zeiss). Images were taken using the Zen 3.8 Pro software with *Z*-stack every 0.5 µm with a range of 3 µm (seven slices). Images were analysed using Fiji[53].

## Protein secretion assay

After the cells were cultured using the pulsed *flhDC* induction set-up, a 1.9-ml aliquot was taken, and cells were collected by centrifugation at 13,000 × *g* for 5 min. Then, 1 ml of the supernatant was transferred into a fresh tube, the remaining supernatant was discarded and the cell

pellets were resuspended in 1 ml of double-distilled water. Proteins were precipitated from the supernatant and pellet fractions using 10% (v/v) TCA, followed by a 30-min incubation step on ice and centrifugation at 20,000 × g for 30 min. Protein pellets were washed with ice-cold acetone and air-dried. Samples were adjusted to 20 OD units μl⁻¹, and 200 OD units were analysed under denaturing conditions using SDS-PAGE. Immunoblotting was performed using primary α-FliC (Difco, catalogue number 228241 *Salmonella* H Antiserum I, 1:5,000 in 1 × TBS-T), α-FliD (gift from T. Minamino, 1:10,000 in 1 × TBS-T) or α-FlgK (gift from T. Minamino, 1:10,000 in 1 × TBS-T) antibodies. Proteins were detected using secondary α-rabbit antibodies conjugated to horseradish peroxidase (Bio-Rad Immun-Star Goat Anti-Rabbit-HRP Conjugate, catalogue number 170-5046, 1:20,000 in 1 × TBS-T). The relative amounts of secreted and cellular proteins were determined by normalization to the housekeeping protein DnaK using the Image Lab software (Bio-Rad). DnaK was detected using primary α-DnaK (Abcam) antibodies and secondary α-mouse antibodies conjugated to horseradish peroxidase (Bio-Rad Immun-Star Goat Anti-Mouse-HRP Conjugate, catalogue number 170-5047, 1:20,000 in 1 × TBS-T).

## Statistical analyses
Statistical analyses were performed using GraphPad Prism 10 (GraphPad Software), and values of $P < 0.05$ were considered statistically significant.

## Cryo-ET sample preparation of the *S. enterica* FliD cap
An overnight culture of strain EM8327 (Supplementary Table 2) in LB was diluted 1:100 in 10 ml of LB and incubated for 1.5 h. AnTc was added to induce flagellar biosynthesis, and the cells were further incubated for 45 min. Cells were centrifuged at 2,550 × g for 2 min and resuspended in remaining LB. A 5% BSA solution was used to resuspend the pellet and left at 4 °C for 30 min, after which the sample was centrifuged at 6,000 × g for 10 min, and the pellet was resuspended in 200 μl of 50 mM HEPES and 100 mM NaCl pH 7 buffer. This was followed by an additional centrifugation step under the same conditions and resuspension in 35 μl buffer. Colloidal gold solution (1 ml, 5 nM) was centrifuged for 10 min at 6,000 × g. The pipette tip was cut off to allow larger cells to pass through onto the grid undamaged, and 10 μl of sample was mixed with 10 μl of colloidal gold solution before loading 5 μl of the mixture onto C-flat holey carbon films 3.5/1200 mesh (EMS) using a Leica EM GP plunge freezer. Cryo-EM grids were negative glow discharged for 30 s and the double blotting strategy was used, in which 5 μl of sample was loaded onto the grid for 2 min at 80% humidity and 4 °C chamber conditions, back blotted, loaded once more for 2 min, back blotted and loaded for 1 min, and back blotted for 6.5 s before plunging into liquid ethane to freeze the grid.

## Cryo-ET data collection and processing of the *S. enterica* FliD cap
Cryo-EM tomography data were collected using a Titan Krios TEM (Thermo Fisher) operated at 300 kV and equipped with a Falcon IV camera. In total, 68 tomograms were collected using the EPU software (Thermo Fisher) in linear mode, with a pixel size of 3.6 Å pix⁻¹, with a total dose of 86 e⁻ Å⁻² spread across 35 tilts with 10 fractions each in 3 degree increments with a range of −51 to +51 degrees. The defocus range used for data collection was approximately −2 μm to −8 μm.

Preprocessing was performed using the WarpEM software package v1.0.9 and alignment was done using IMOD v4.11.11 (refs. 62,63). Particle picking was performed manually on binned and deconvoluted tomograms using the 3Dmod software. Coordinates were transferred to a .star file, and sub-tomogram reconstruction was performed in WarpEM and imported to Relion v4.0 with 252 sub-tomograms[63–65].

The flagellum filament model (EMD-9896) was chosen as a starting model, low-pass filtered to 60 Å and subjected to 3D refinement in Relion v4.0, with C1 symmetry, generating de novo density for the tip[65].

Further 3D classification and CTF refinement in the WarpEM software did not further increase the map resolution, presumably because of the limited number of sub-tomograms.

## *C. jejuni* strain construction and cultivation
A *C. jejuni* minicell (Δ*flhG* Δ*flaAB*) strain was constructed as described previously[66,67]. Briefly, *aphA-rpsL*^WT cassettes flanked by ~500 bp overhangs with homology to the targeted chromosomal loci and EcoRI sites at the 5′ and 3′ termini were synthesized by 'splicing by overlap extension' PCR. Linear DNA fragments were methylated at their ecoRI sites with ecoRI methyltransferase (New England Biolabs) and transformed into *C. jejuni* using the biphasic method[68]. Transformants were selected for on Müller-Hinton (MH) agar supplemented with 50 μg ml⁻¹ kanamycin. Replacement of the *aphA-rpsL*^WT with the desired mutation was achieved using the same method, but with transformants being selected for on MH agar supplemented with 2 mg ml⁻¹ streptomycin sulfate. Kanamycin-sensitive, streptomycin-resistant transformants were single-colony purified and checked by Sanger sequencing (Source Bioscience). For the minicell background, in-frame deletion of *flhG* leaves the first and last 20 codons intact, while the Δ*flaAB* allele spans from 20 base pairs upstream of the *flaA* translational start site to codon 548 of *flaB*.

## Cryo-EM sample preparation of *C. jejuni* minicells
*C. jejuni* Δ*flhG* Δ*flaAB* cells were grown on MH plates and resuspended in PBS buffer (137 mM NaCl, 2.7 mM KCl, 10 mM Na₂HPO₄, 1.8 mM KH₂PO₄, pH 7.4). Cells were spun at 1,500 × g for 20 min to pellet whole cells. The minicell-enriched supernatant was removed and spun in a tabletop microcentrifuge at 15,000 × g for 5 min to pellet the minicells. The pellet was then resuspended to a theoretical OD₆₀₀ of ~15.

Minicells were vitrified on QUANTIFOIL R0.6/1 or R1.2/1.3 holey carbon grids (Quantifoil Micro Tools) using a Vitrobot Mark IV (Thermo Fisher Scientific).

## Single-particle cryo-EM data collection and processing of the *C. jejuni* cap−HFJ complex
Particle coordinates were found using a crYOLO[69] model derived from manually picked particles from 2,718 micrographs and a box size of 180 nm. Processing was done in RELION 4.0 (ref. 64). CTF correction was performed using CTFFIND-4.1 (ref. 70). A total of 79,106 particles was extracted from 42,988 micrographs (2.2 Å pix⁻¹). After several rounds of 2D classification and a 3D classification with a 500-Å-diameter mask, particles were re-centred at the tip of the axial structure and re-extracted. Additional rounds of 2D classifications and a 3D classification with a 400-Å-diameter mask with no symmetry imposed and post-processing to yield the final map. For the atomic model of the *C. jejuni* cap−junction complex, monomer structures of *C. jejuni* FliD, FlgK and FlgL were generated using AlphaFold2 (ref. 48). All subunits were manually fitted into the reconstructed maps as rigid bodies, including 11 FlgK, 11 FlgL and 5 FliD. The entire complex was refined with secondary structure restraints in ISOLDE[49]. PHENIX[51] and Coot[52] were used to refine further and validate in the same way we refined the above *Salmonella* models. The *Campylobacter* HFJ−cap structure showed an average Q-score of 0.2, which is expected given its lower resolution.

## Reporting summary
Further information on research design is available in the Nature Portfolio Reporting Summary linked to this article.

## Data availability
The cryo-ET map of the *S. enterica* flagellar tip has been deposited in the Electron Microscopy Data Bank (EMDB). The coordinates and EM maps, including the *S. enterica* cap−filament complex, the *S. enterica* HFJ and the *C. jejuni* cap−HFJ complex, have been deposited in the PDB

and EMDB databases with the following accession codes: the cryo-ET map of *S. enterica* flagellar tip, EMD-51555; the *S. enterica* cap–filament complex, PDB: 9GNZ and EMDB: EMD-51486; the *S. enterica* HFJ, PDB: 9GO6 and EMDB: EMD-51493; and the *C. jejuni* cap–HFJ complex, PDB: 9GSX and EMDB: EMD-51557. Source data are provided with this paper.

## Code availability

The code used for the analysis of the single-cell tracking data is available via GitHub at https://github.com/SalmoLab/FlagellumStructure_NatMicro2025 (ref. 23).

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

## Acknowledgements

We acknowledge the support of the European Molecular Biology Organization (EMBO) through an EMBO Scientific Exchange Grant for the research stay of R.E. at the Bergeron laboratory. K.Q. is supported by a PhD studentship from the China Scholarship Council. M.E. acknowledges funding from the European Research Council under the European Union's Horizon 2020 research and innovation programme (grant agreement number 864971) and from the Max Planck Society as a Max Planck Fellow. J.R.C.B. acknowledges funding from the Biotechnology and Biological Sciences Research Council (BB/R009759/2) and Human Frontier Science Program (RGY0080/2021). We thank members of the Erhardt, Bergeron and Beeby laboratories for helpful discussions. We thank P. F. Popp for providing the code for single-cell tracking analysis. We thank T. Minamino (Osaka University) for providing antibodies, C. Goosmann (Max Planck Institute for Infection Biology) for TEM grid preparations and observation of purified flagellum samples for protocol optimizations, and C. Odendall and J. Wanford for help with access to Cat-2 laboratories. Cryo-EM grids were screened at the Imperial College London cryo-EM facility (funded by BBSRC grant BB/V019732/1), and data were collected at the LonCEM facility; we acknowledge P. Simpson and N. Cronin, respectively, for support.

## Author contributions

J.R.C.B., M.E. and M.B. conceptualized and supervised the research project and obtained funding for the study. R.E. prepared the samples of purified short flagella of *S. enterica* and reconstructed the EM map of the FliD cap complex of *S. enterica* with the help of K.Q. K.Q. prepared the EM grids, and collected and processed the EM data. K.Q. reconstructed the EM map of the cap complex and HFJ complex of *S. enterica* and built the corresponding atomic models. R.E. and J.S. generated chromosomal *S. enterica* FliD mutants, and performed and analysed motility assays, secretion assays and fluorescent microscopy experiments of FliD mutants. J.S. performed leakage assays. R.E. generated chromosomal *S. enterica* FlgKL mutants and performed and analysed filament shearing assays. N.S.A. prepared samples for cryo-ET of intact *S. enterica* cells and contributed to EM grid preparation, EM data

acquisition and EM data processing, with help from D.M. T.D., E.J.C., N.G.-R., J.H., E.S. and M.B. designed and conducted the experiments, including generating the *C. jejuni* minicell strain, collecting and processing the cryo-EM data of *C. jejuni* minicells and obtaining the map of the cap–HFJ complex. K.Q. built the atomic models of the cap–HFJ complex of *C. jejuni*. R.E. and K.Q. wrote the first draft of the paper and prepared the figures, and K.Q. prepared the videos. M.E. and J.R.C.B. reviewed and edited the paper. All authors reviewed the results and approved the final version of the paper. J.S. and N.S.A. contributed equally as joint second authors.

## Competing interests

The authors declare no competing interests.

## Additional information

**Extended data** is available for this paper at https://doi.org/10.1038/s41564-025-02037-0.

**Correspondence and requests for materials** should be addressed to Marc Erhardt or Julien R. C. Bergeron.

**Extended Data Table 1 | Surface area between FliD and its adjacent FliCs**

| FliD # | Interface area between FliD and FliCs (Å²) | | | Total area (Å²) |
|---|---|---|---|---|
| | **+5** | **0** | **+6** | |
| 1 | 1790 | 796 | 683 | 3269 |
| 2 | 1684 | 690 | 841 | 3215 |
| 3 | 1768 | 780 | 645 | 3193 |
| 4 | 470 | 822 | - | 1292 |
| 5 | 1728 | 627 | 699 | 3054 |

**Extended Data Table 2 | Surface area between FlgK or FlgL and their interaction partners**

| Subject | Mode | Interface area between FlgK/FlgL and adjacent proteins (Å²) | | | | | |
|---------|------|-------|-------|-------|-------|-------|-------|
| | | **+5** | **+11** | **+6** | **−5** | **−11** | **−6** |
| **FlgK** | **1** | 2220.7 (FlgK) | 1688.7 (FlgL) | 456.4 (FlgL) | 2214.9 (FlgK) | 1163.8 (FlgE) | 1045.4 (FlgE) |
| | **2** | 2214.9 (FlgK) | 1568.6 (FlgL) | 992.1 (FlgK) | 1400.5 (FlgE) | 1003.0 (FlgE) | 944.4 (FlgE) |
| | **3** | 1552.9 (FlgL) | 1641.6 (FlgL) | 251.8 (FlgL) | 2188.9 (FlgK) | 1627.6 (FlgE) | 992.1 (FlgK) |
| **FlgL** | **1** | 1507.2 (FlgL) | 1881.8 (FliC) | 410.6 (FliC) | 1661.9 (FlgL) | 1688.7 (FlgK) | 474.3 (FlgK) |
| | **2** | 1661.9 (FlgL) | 1743.4 (FliC) | 203.6 (FlgL) | 1552.9 (FlgK) | 1568.6 (FlgK) | 456.4 (FlgK) |
| | **3** | 1263.7 (FliC) | 1809.0 (FliC) | 513.5 (FliC) | 1240.4 (FlgL) | 1641.6 (FlgK) | 203.6 (FlgL) |

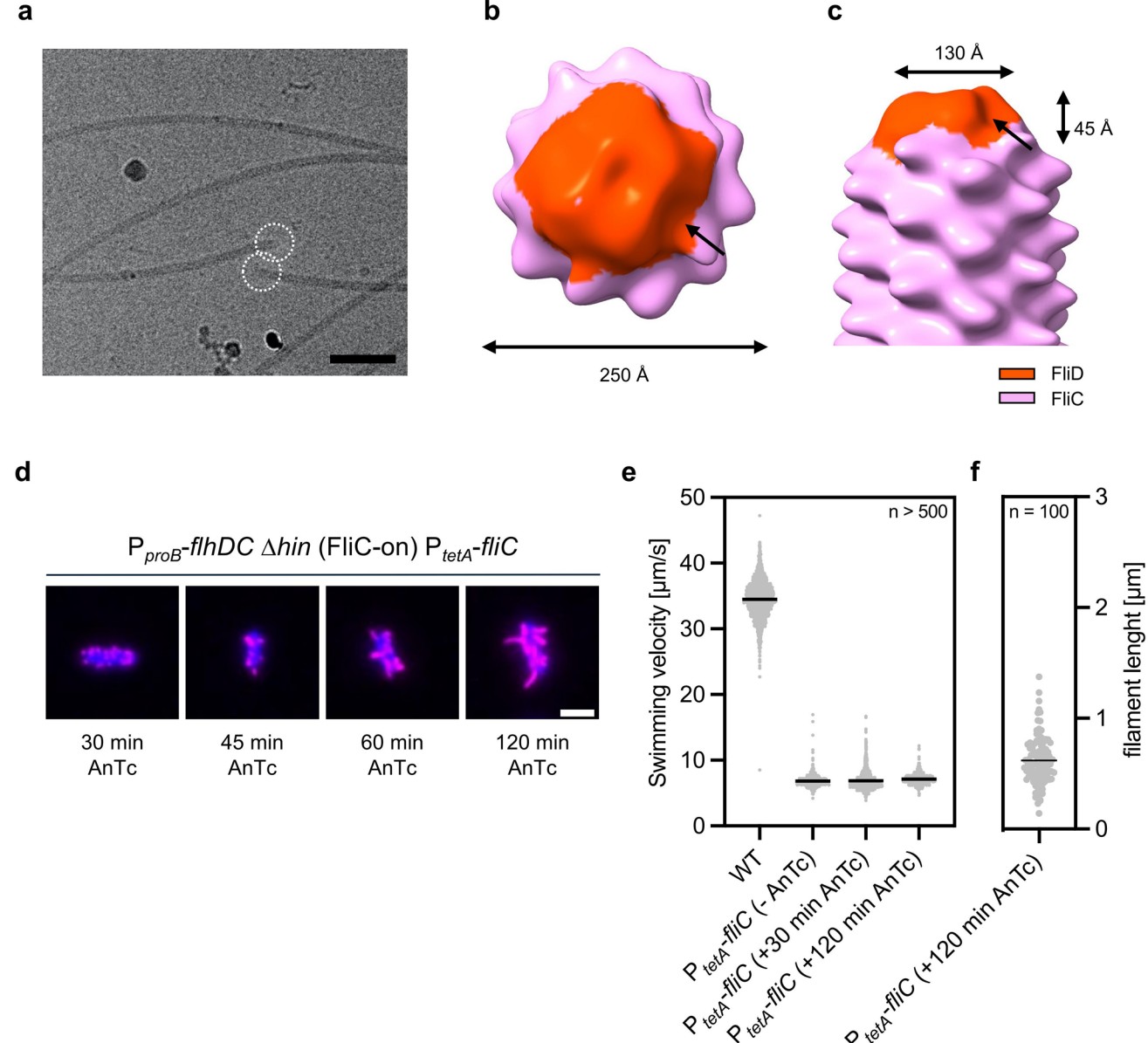

**Extended Data Fig. 1 | Cryo-ET of filament tips and characteristics of the short flagella mutant. a**, *S. enterica* flagella in cryo-EM. Scale bar: 100 nm. **b**, Top view of flagellar tip with the diameter of filament. **c**, Side view of flagellar tip with the dimensions of the FliD cap. **d**, Representative fluorescence microscopy images of *S. enterica* EM16009 that was used for short flagella purification. Filaments (FliC) were immunostained with anti-FliC primary antibody and anti-rabbit coupled to GFP after inducing flagellin production with anhydrotetracycline (AnTc) for 30, 45, 60, or 120 min. DNA counterstained with DAPI. Colour of the GFP channel changed to magenta. Scale bar: 2 μm. One biological replicate performed.

**e**, Single-cell swimming velocities of the hyperflagellated wildtype (WT, EM13434) and the short flagella mutant strain (P$_{tetA}$:*fliC*, EM16009) in absence or, when required, presence of AnTc to induce FliC production. Data points represent the swimming velocity of individual trajectories, the horizontal bar represents the mean swimming velocity of at least 500 cells per strain and condition. Experiment was performed as one biological replicate. **f**, Quantification of the filament length in μm of the short flagella mutant EM16009 after 120 min of AnTc induction. Data points represent individual filaments, black line indicates the mean of n = 100 individual filaments of one biological replicate.

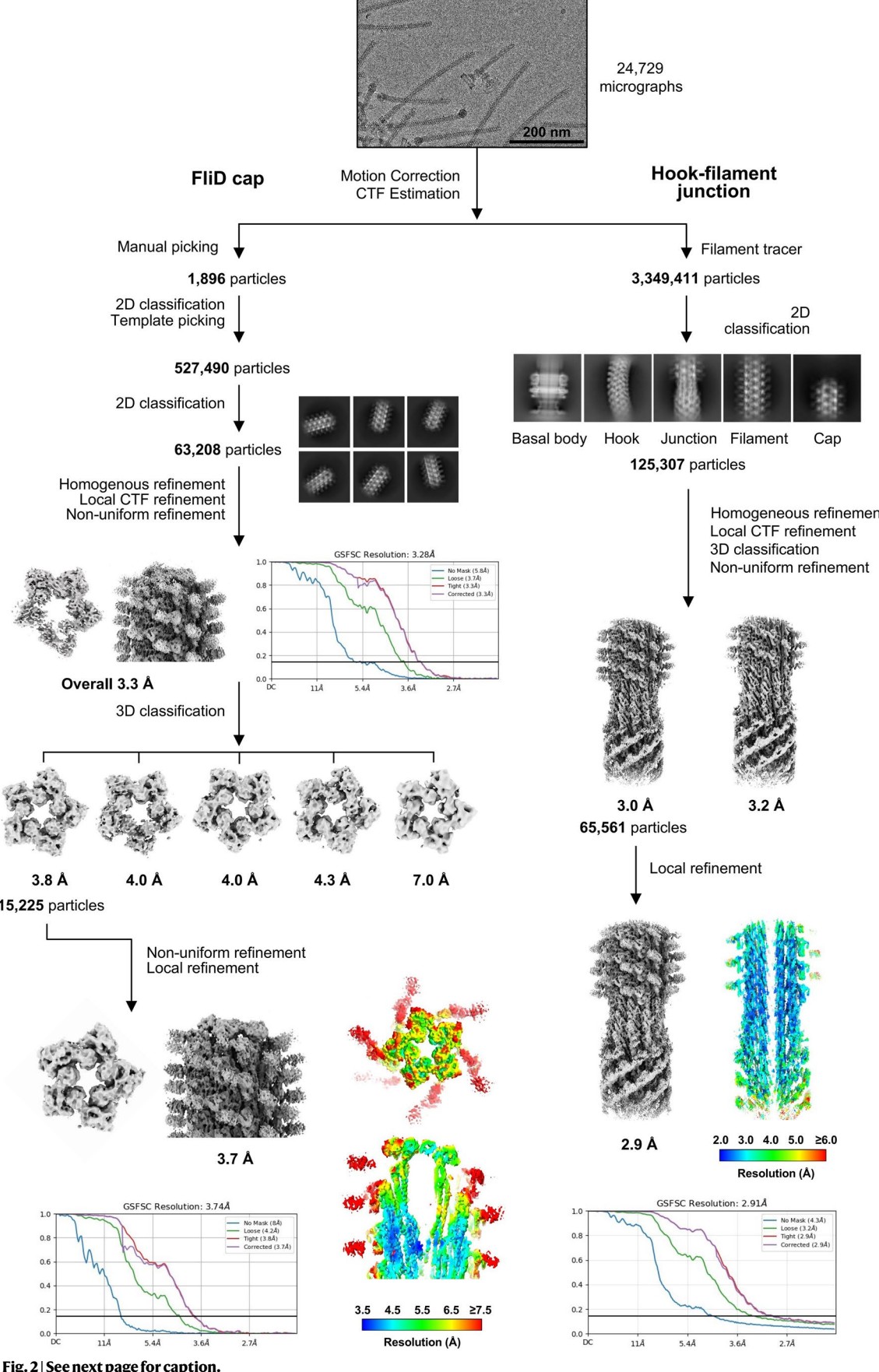

**Extended Data Fig. 2 | See next page for caption.**

**Extended Data Fig. 2 | Cryo-EM data processing pipeline.** Flowchart of data collection and processing pipeline in cryoSPARC that resulted in the final *S. enterica* HFJ and cap complex cryo-EM structure. FliD cap: 1,896 particles were manually picked to generate a template. Template-based picking resulted in 527,490 particles from 24,729 micrographs. After 2D classification, 63,208 particles were used to generate and refine an initial model, obtaining a 3.3 Å resolution map. 3D classification was performed to solve heterogeneity in the sample. The largest class containing 15,225 particles was further refined, obtaining a 3.7 Å resolution map and cryo-EM density maps coloured by local resolution (in Å) of the top-view and cross-section of the FliD cap in complex with the filament estimated in cryoSPARC shown. Flowchart includes Gold-Standard Fourier Shell Correlations of the initial map with an estimated global resolution of 3.3 Å at FSC = 0.143 and of the final map with an estimated resolution of 3.7 Å at FSC = 0.143. Hook-Filament Junction: The use of Filament Tracer led to an initial set of 3,349,411 particles from the same 24,729 micrographs. After 2D classification, 125,307 particles were used to generate an initial map. After 3D classification and refinements of the largest class containing 65,561 particles, the map reached a resolution of 3.0 Å. Local refinement further improved resolution to 2.9 Å and cryo-EM density map coloured by local resolution (in Å) of the cross-section of the HFJ in complex with the hook and filament estimated in cryoSPARC shown. Flowchart includes Gold-Standard Fourier Shell Correlation of the final map with an estimated global resolution of 2.9 Å at FSC = 0.143.

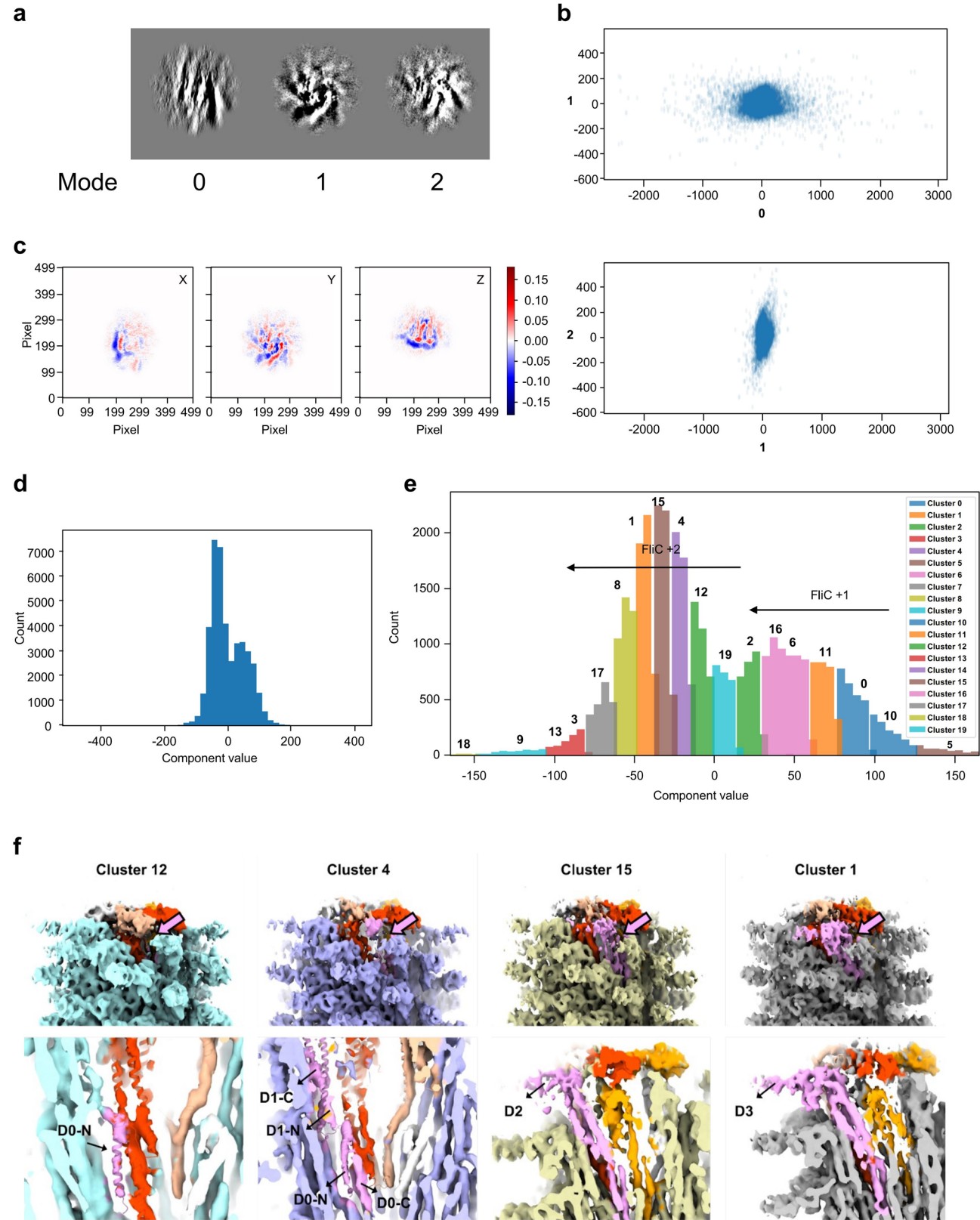

**Extended Data Fig. 3 | 3D variability analysis of FliD-FliC complex. a**, Projection of each variability modes. **b**, Scatter plots of reaction coordinate distribution of particles between adjacent pairs of components. **c**, Projection of variability mode 1 from x, y, z axis. **d**, The distribution of latent coordinates for the component in **c**. **e**, Histogram of clusters of latent coordinates. Clusters 10 to 2 represent the first flagellin being incorporated, while clusters 19 to 17 represent the second flagellin. **f**, Volume map that are reconstructed from clusters 12, 4, 15, 1 in **e**.

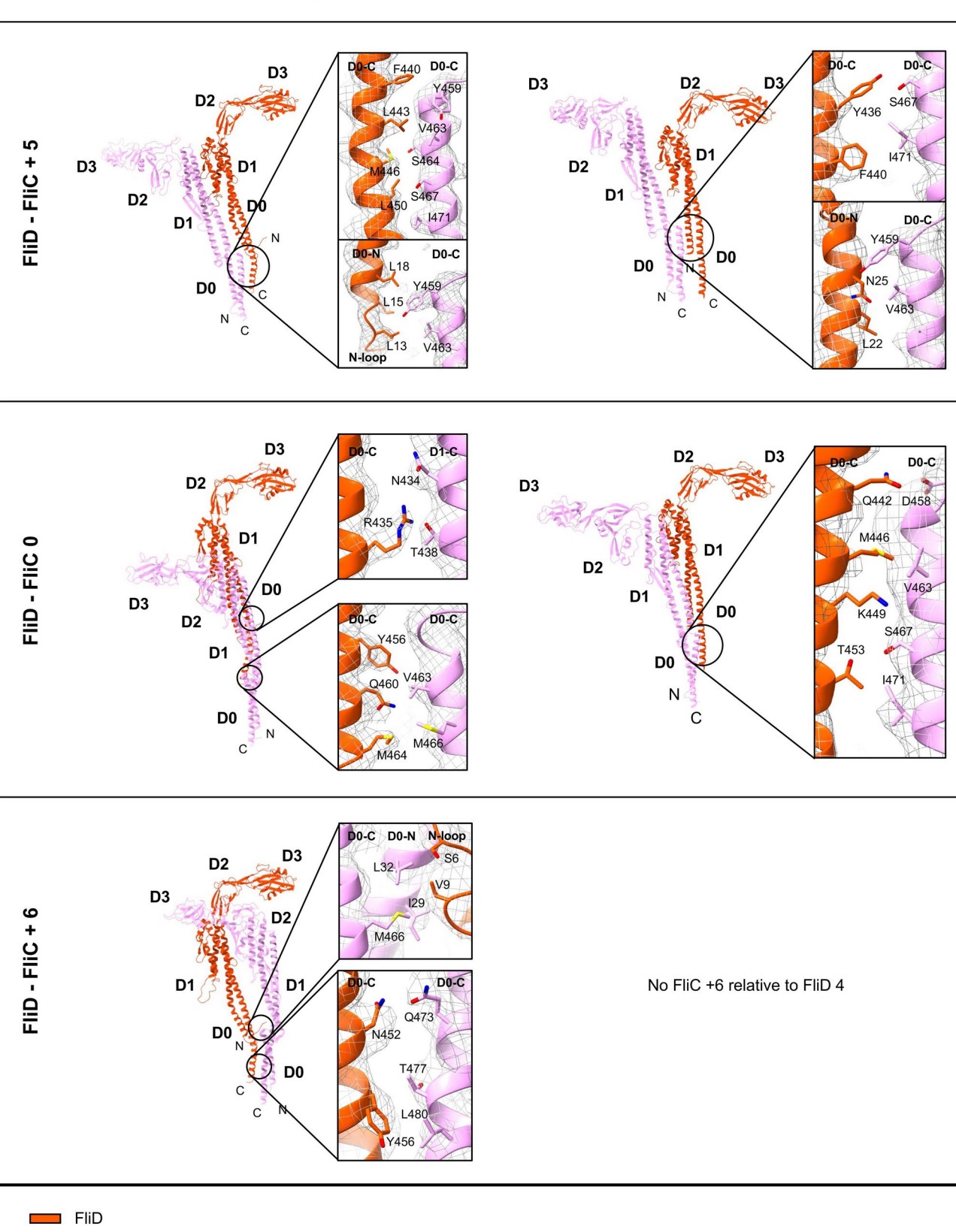

**Extended Data Fig. 4 | Details of the FliC-FliD interactions at their D0-D1 domains.** The interface between each FliD and FliC at position +5, 0, +6 are displayed and residues that are potentially interacting are displayed. FliD 1, 2, 3, 5 are grouped as they adopt similar interactions with adjacent FliC.

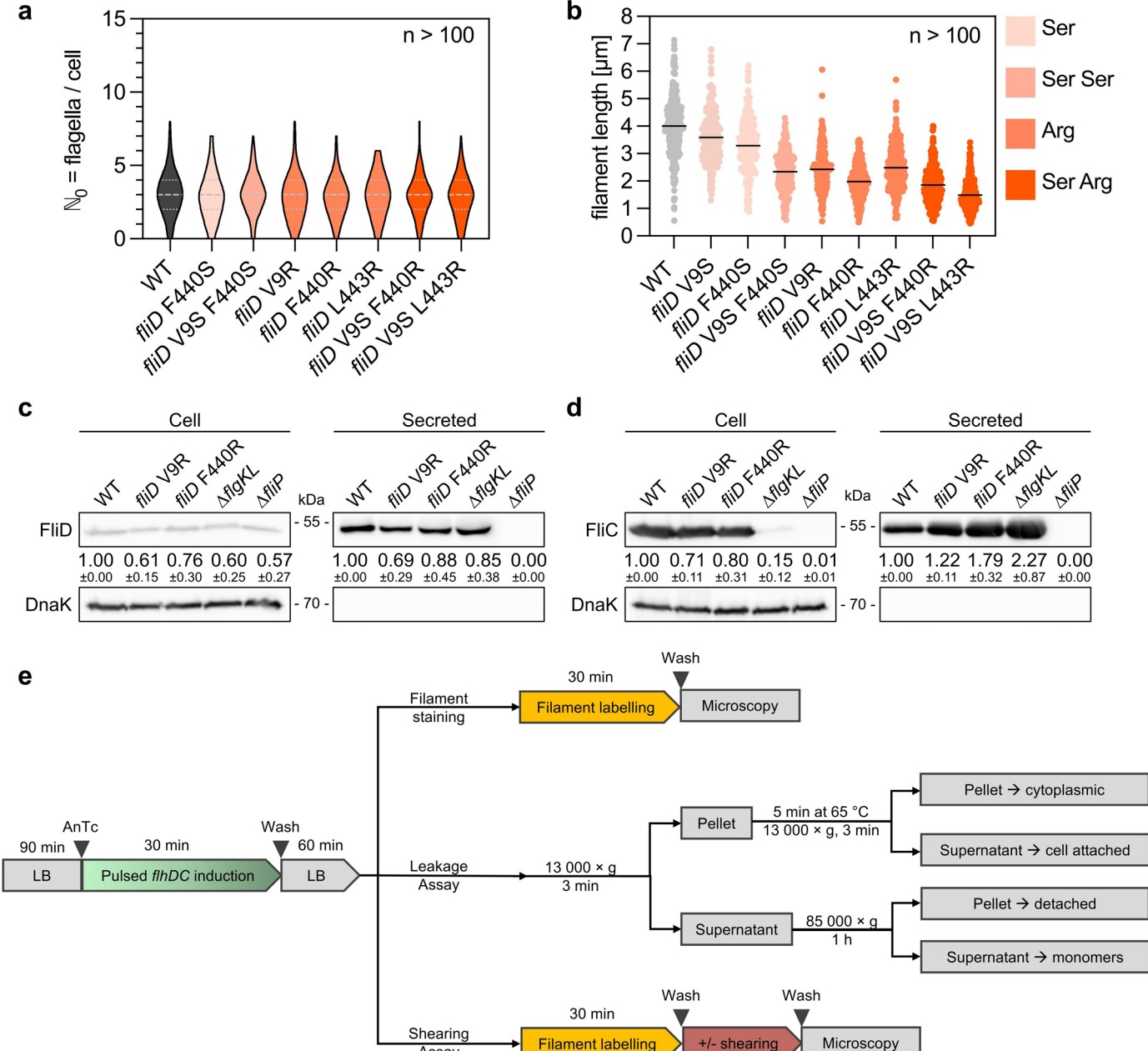

**Extended Data Fig. 5 | Comparative analysis of filament characteristics and protein secretion in wild-type and FliD mutant strains. a,** Quantification of the number of flagella per cell in the wild-type and FliD mutants using the pulsed *flhDC* induction setup. The number of flagella per cell ($\mathbb{N}_0$) was determined for n > 100 individual bacteria per strain for one biological replicate. Violin plots represent the distribution of the data including the median (dashed line) and quartiles (dotted lines). **b,** Quantification of the filament length in µm in the wild-type and FliD mutants using the pulsed *flhDC* induction setup. The filament length was determined for n > 100 individual filaments per strain for one biological replicate. All data points shown, black lines indicate the means. Ser, single substitution to serine; Ser Ser, double substitution to serine; Arg, single

substitution to arginine; Ser Arg, double substitution to serine and arginine. **c,d,** Immunoblotting of the cellular and secreted fractions using the pulsed *flhDC* induction setup of wild-type, the FliD mutants (V9R and F440R), a Δ*flgKL* strain and a secretion-deficient Δ*fliP* mutant using **c,** anti-FliC or **d,** anti-FliD antibodies. Relative secreted FliC or FliD levels report mean ± standard deviation, n = 3. DnaK serves as a loading and lysis control and was used to normalize the protein levels. **e,** Pulsed *flhDC* induction setup used to analyse filament number and filament length with fluorescence microscopy, and perform secretion, leakage and shearing assays. For secretion assays, the first Pellet and Supernatant samples of the Leakage assay were directly used for TCA precipitation and SDS-PAGE analysis, without the additional steps.

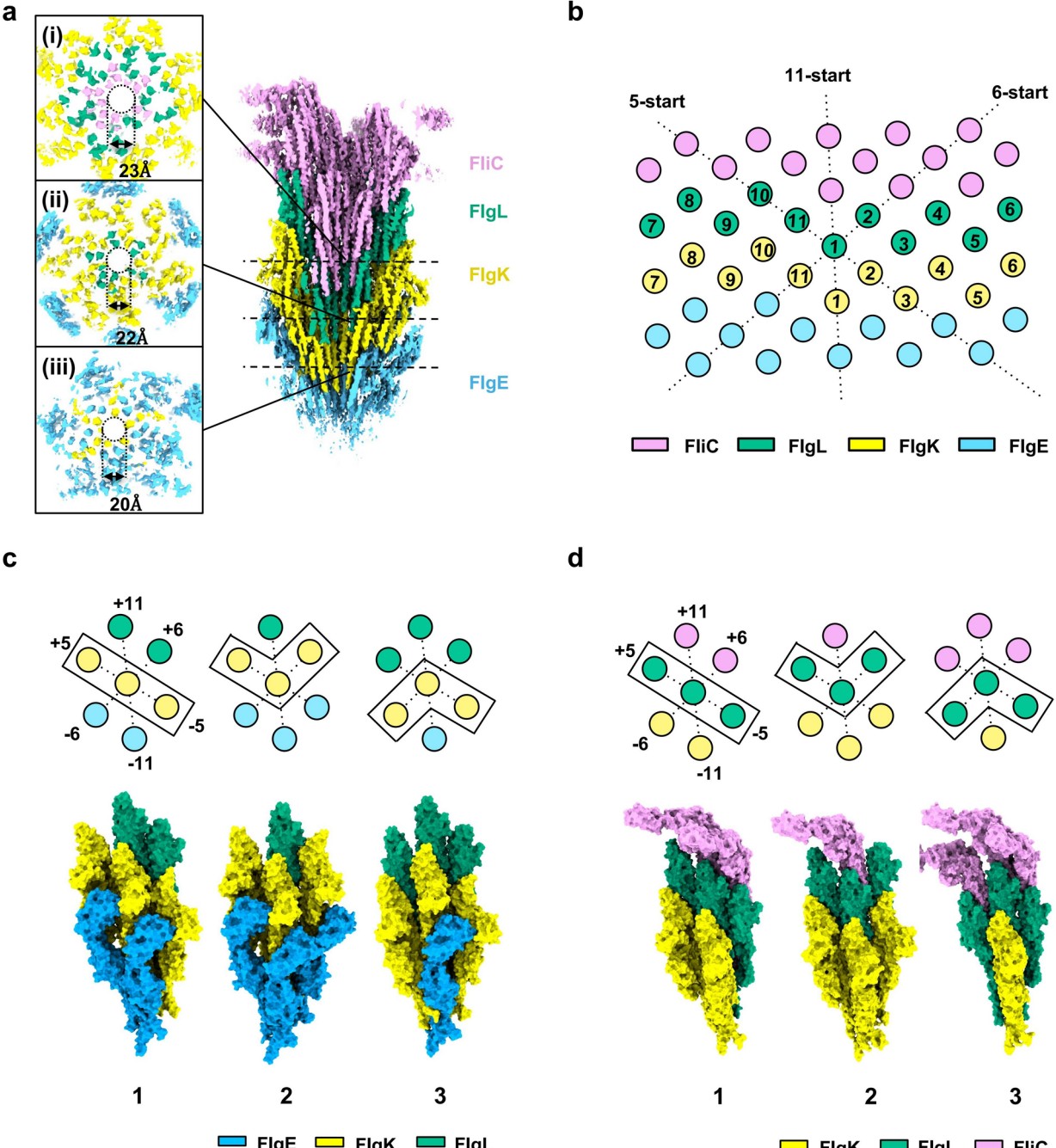

**Extended Data Fig. 6 | Details of the interactions of FlgK and FlgL with neighbouring subunits. a**, Cross-section view of the HFJ in complex with the hook and the filament (right) with labelled width of the secretion channel at the bottom of the FliC filament (i), the FlgL layer (ii) and the FlgK layer (iii).

**b**, The arrangement of components in HFJ in the symmetry lattice along 5-start, 6-start, 11-start axes. **c**, FlgK interacts with 6 adjacent proteins in 3 different modes. **d**, FlgL interacts with 6 adjacent proteins in 3 different modes.

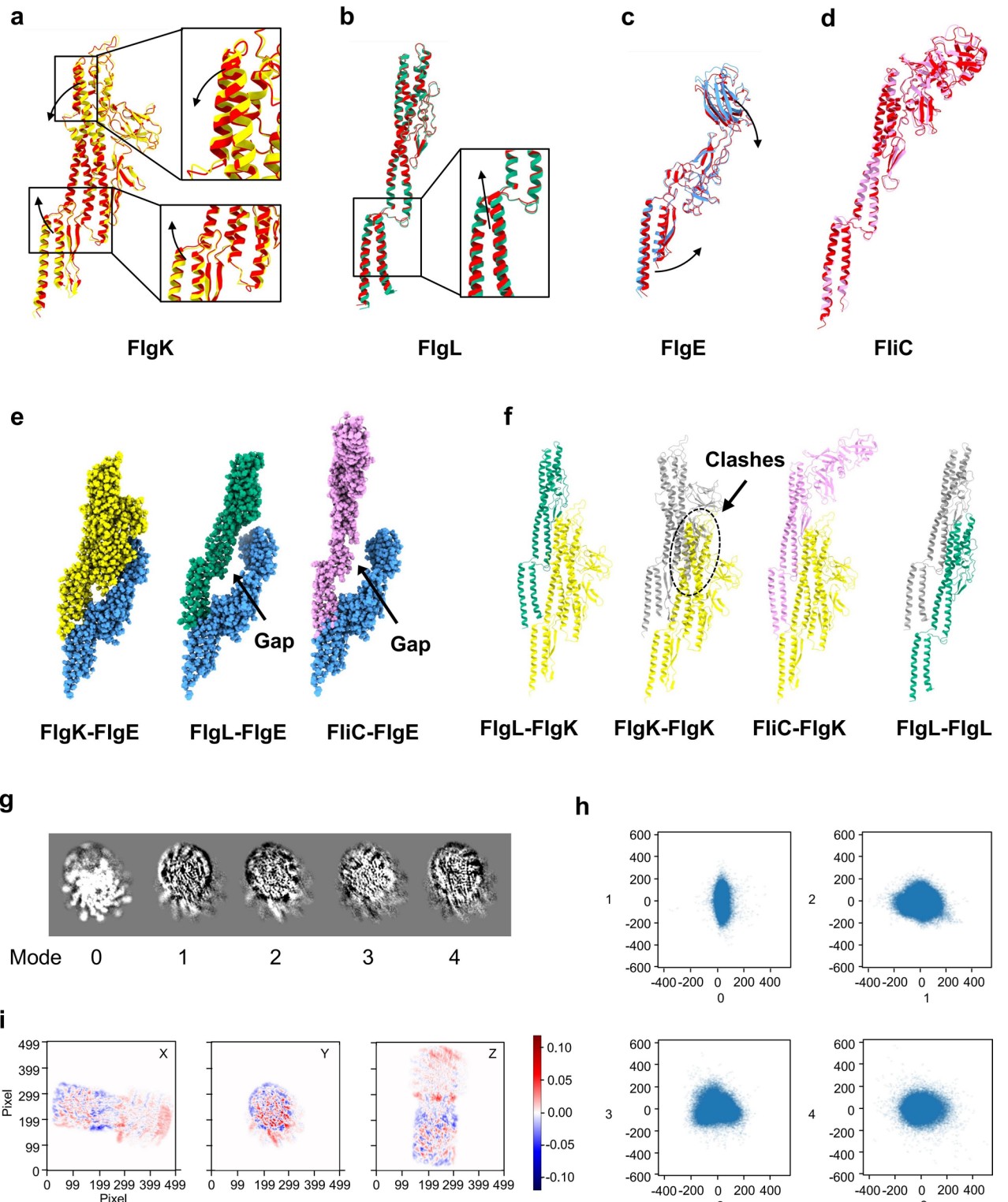

**Extended Data Fig. 7 | FlgK acts as an elastic buffer, preventing stress propagation from the hook to the filament. a-d**, Structural superimposition of the FlgE-FlgK-FlgL-FliC protofilament components in their extended and compressed states. The compressed conformations are shown in red. Subunits of FlgK (**a**), FlgL (**b**), FlgE (**c**), and FliC (**d**) in both states are overlaid to illustrate structural changes. Arrows indicate the transition from the extended to the compressed state. **e**, The connection to the FlgE hook is disrupted when FlgK is substituted with FlgL and FliC. The gap of disconnection is highlighted by arrows. **f**, Revealing the assembly of HFJ and FliC filament by molecular substitution. Clashes in FlgK-FlgK complex are highlighted. **g**, Projection of each variability modes. **h**, Scatter plots of reaction coordinate distribution of particles between components. **i**, Projection of variability mode 2 from x, y, z axis.

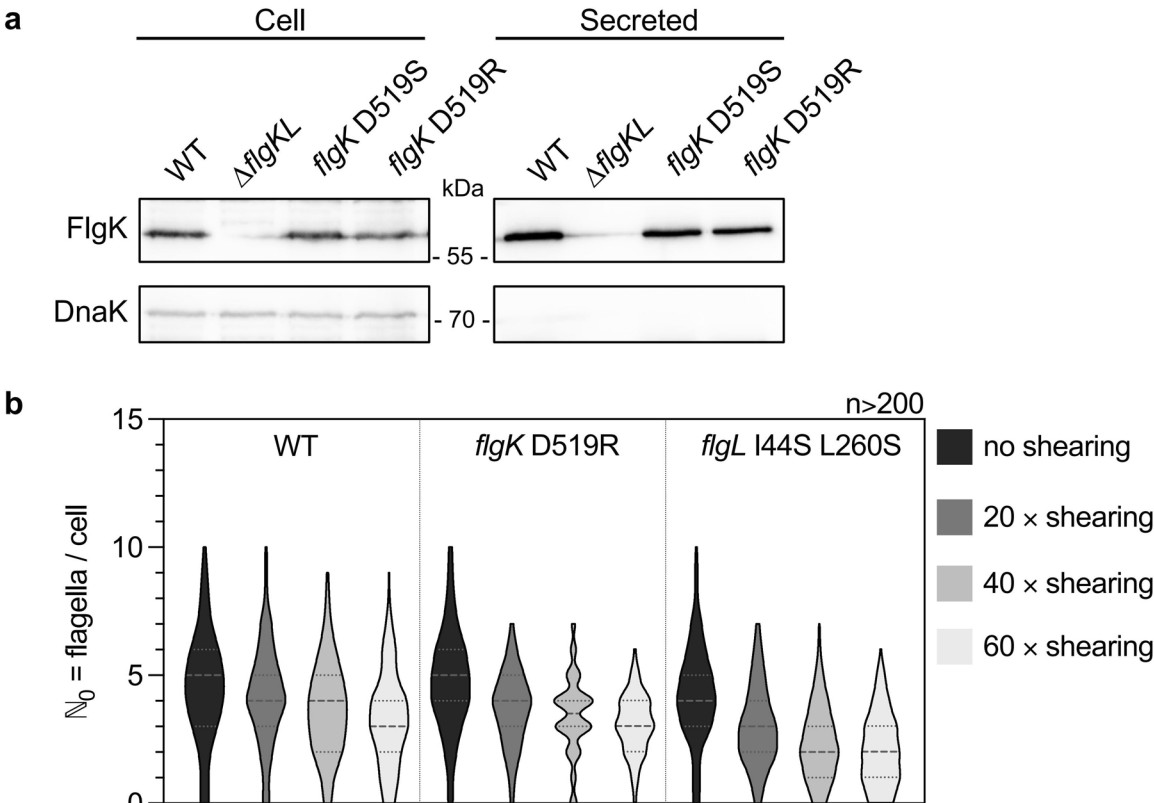

**Extended Data Fig. 8 | FlgK mutant variants are secreted in comparable amounts to WT FlgK and flagella of FlgKL interface mutants are more prone to break. a,** Immunoblotting of the cellular and secreted fractions using the pulsed *flhDC* induction setup of wild-type, a Δ*flgKL* control strain, the FlgK mutants (D519S and D519R) using anti-FlgK antibody. DnaK serves as a loading and lysis control. One biological replicate performed. **b,** Quantification of the number of flagella per cell using the pulsed *flhDC* induction setup in the wild-type and the FlgK D519R and FlgL I44S L260S mutants without shearing and after 20 ×, 40 × or 60 × shearing. Filaments (FliC T237C) were labelled with Dylight555 Maleimide and DNA counterstained with DAPI after pulsed *flhDC* induction. The number of flagella per cell ($\mathbb{N}_0$) was determined for n > 200 individual bacteria per strain and condition for one biological replicate. Violin plots represent the distribution of the data including the median (dashed line) and quartiles (dotted lines).

Marc Erhardt

# Reporting Summary

## Statistics

For all statistical analyses, confirm that the following items are present in the figure legend, table legend, main text, or Methods section.

| n/a | Confirmed | |
|---|---|---|
| ☐ | ☒ | The exact sample size (*n*) for each experimental group/condition, given as a discrete number and unit of measurement |
| ☐ | ☒ | A statement on whether measurements were taken from distinct samples or whether the same sample was measured repeatedly |
| ☒ | ☐ | The statistical test(s) used AND whether they are one- or two-sided<br>*Only common tests should be described solely by name; describe more complex techniques in the Methods section.* |
| ☒ | ☐ | A description of all covariates tested |
| ☒ | ☐ | A description of any assumptions or corrections, such as tests of normality and adjustment for multiple comparisons |
| ☐ | ☒ | A full description of the statistical parameters including central tendency (e.g. means) or other basic estimates (e.g. regression coefficient) AND variation (e.g. standard deviation) or associated estimates of uncertainty (e.g. confidence intervals) |
| ☐ | ☒ | For null hypothesis testing, the test statistic (e.g. *F*, *t*, *r*) with confidence intervals, effect sizes, degrees of freedom and *P* value noted<br>*Give P values as exact values whenever suitable.* |
| ☒ | ☐ | For Bayesian analysis, information on the choice of priors and Markov chain Monte Carlo settings |
| ☒ | ☐ | For hierarchical and complex designs, identification of the appropriate level for tests and full reporting of outcomes |
| ☒ | ☐ | Estimates of effect sizes (e.g. Cohen's *d*, Pearson's *r*), indicating how they were calculated |

*Our web collection on statistics for biologists contains articles on many of the points above.*

## Software and code

Policy information about availability of computer code

| | |
|---|---|
| Data collection | EPU 3 (Thermo Fisher) was used for cryo-EM data collection |
| Data analysis | Cryo-EM data was processed with CryoSPARC v4.3. Atomic models were generated with AlphaFold3, and the structures were refined with Phenix. Structural figures were generated with Pymol or ChimeraX. |

For manuscripts utilizing custom algorithms or software that are central to the research but not yet described in published literature, software must be made available to editors and reviewers. We strongly encourage code deposition in a community repository (e.g. GitHub). See the Nature Portfolio guidelines for submitting code & software for further information.

## Data

Policy information about availability of data

All manuscripts must include a data availability statement. This statement should provide the following information, where applicable:
- Accession codes, unique identifiers, or web links for publicly available datasets
- A description of any restrictions on data availability
- For clinical datasets or third party data, please ensure that the statement adheres to our policy

The Cryo-ET map of S. enterica flagellar tip has been deposited in the EMDB database. The coordinates and EM maps including the S. enterica cap-filament complex, the S. enterica HFJ, and the C. jejuni cap-HFJ complex, have been deposited in the PDB and EMDB databases with the following accession code: the cryo-ET map of

S. enterica flagellar tip, EMD-51555; the S. enterica cap-filament complex, PDB: 9GNZ, EMDB: EMD-51486; S. enterica HFJ, PDB: 9GO6, EMDB: EMD-51493; C. jejuni cap-HFJ complex, PDB: 9GSX, EMDB: EMD-51557.

# Research involving human participants, their data, or biological material

Policy information about studies with human participants or human data. See also policy information about sex, gender (identity/presentation), and sexual orientation and race, ethnicity and racism.

| | |
|---|---|
| Reporting on sex and gender | N.A |
| Reporting on race, ethnicity, or other socially relevant groupings | N/A |
| Population characteristics | N/A |
| Recruitment | N/A |
| Ethics oversight | N/A |

Note that full information on the approval of the study protocol must also be provided in the manuscript.

# Field-specific reporting

Please select the one below that is the best fit for your research. If you are not sure, read the appropriate sections before making your selection.

☒ Life sciences    ☐ Behavioural & social sciences    ☐ Ecological, evolutionary & environmental sciences

For a reference copy of the document with all sections, see nature.com/documents/nr-reporting-summary-flat.pdf

# Life sciences study design

All studies must disclose on these points even when the disclosure is negative.

| | |
|---|---|
| Sample size | Filament-cap complex: 15,225 particles<br>HFJ complex: 65,561 particles<br>HFJ-cap complex: 15,077 particles<br>These particles were obtained by rounds of 2D/3D classification, and were not pre-established prior to the experiment. |
| Data exclusions | Particles were excluded based on 2D and 3D classification, as described in the manuscript |
| Replication | The structures were obtained through averaging of all the particles for each of the complex. No biological replicates were performed. |
| Randomization | Resolution was estimated using the gold-standard FSC method (FSC=0.143), with the two half-maps generated by random particle distribution. |
| Blinding | Cryo-EM analysis relies on the averaging of 10,000 particles, and no blinding is required. |

# Reporting for specific materials, systems and methods

We require information from authors about some types of materials, experimental systems and methods used in many studies. Here, indicate whether each material, system or method listed is relevant to your study. If you are not sure if a list item applies to your research, read the appropriate section before selecting a response.

## Materials & experimental systems

| n/a | Involved in the study |
|---|---|
| ☐ | ☒ Antibodies |
| ☒ | ☐ Eukaryotic cell lines |
| ☒ | ☐ Palaeontology and archaeology |
| ☒ | ☐ Animals and other organisms |
| ☒ | ☐ Clinical data |
| ☒ | ☐ Dual use research of concern |
| ☒ | ☐ Plants |

## Methods

| n/a | Involved in the study |
|---|---|
| ☒ | ☐ ChIP-seq |
| ☒ | ☐ Flow cytometry |
| ☒ | ☐ MRI-based neuroimaging |

## Antibodies

| | |
|---|---|
| Antibodies used | (1) BD Difco™ Salmonellen H-Antiserum i. - Primary anti-FliC antibody (Cat # 11712894, lot # 1217577)<br>(2) Anti-FliD antibody - gift from Dr Tohru Minamino<br>(3) Anti-FlgK antibody - gift from Dr Tohru Minamino<br>(4) Secondary anti-rabbit antibody Alexa Fluor™ 488 (Invitrogen Cat # A-11034, lot # 1851447)<br>(5) Anti-DnaK antibody (Abcam Cat # ab69617, lot # 103701-2)<br>(6) Immun-Star Goat Anti-Mouse (GAM)-HRP Conjugate (BioRad Cat # 170-5047)<br>(7) Immun-Star Goat Anti-Rabbit (GAR)-HRP Conjugate (BioRad Cat # 170-5046) |
| Validation | (1) The commercially available Difco Salmonella H Antiserum is generally used in tube agglutination tests for the identification of Salmonella by flagellar (H) antigens and has been validated in our laboratory against Salmonella enterica serovar Typhimurium LT2 deletion mutants of FliC.<br>(2) Primary antibody was validated against Salmonella enterica serovar Typhimurium LT2 deletion mutants of FliD.<br>(3) Primary antibody was validated against Salmonella enterica serovar Typhimurium LT2 deletion mutants of FlgK.<br>(4) Immunofluorescence (IF), Flow Cytometry (Flow)<br>(5) WB, ICC/IF, ELISA; Reactivity tested on E. coli lysates, Primary antibody was validated in Salmonella enterica serovar Typhimurium LT2 in our laboratory and produces a single band of the expected size.<br>(6) Commonly used for WB<br>(7) Commonly used for WB |

## Plants

| | |
|---|---|
| Seed stocks | *Report on the source of all seed stocks or other plant material used. If applicable, state the seed stock centre and catalogue number. If plant specimens were collected from the field, describe the collection location, date and sampling procedures.* |
| Novel plant genotypes | *Describe the methods by which all novel plant genotypes were produced. This includes those generated by transgenic approaches, gene editing, chemical/radiation-based mutagenesis and hybridization. For transgenic lines, describe the transformation method, the number of independent lines analyzed and the generation upon which experiments were performed. For gene-edited lines, describe the editor used, the endogenous sequence targeted for editing, the targeting guide RNA sequence (if applicable) and how the editor was applied.* |
| Authentication | *Describe any authentication procedures for each seed stock used or novel genotype generated. Describe any experiments used to assess the effect of a mutation and, where applicable, how potential secondary effects (e.g. second site T-DNA insertions, mosiacism, off-target gene editing) were examined.* |

