## [Peer Review File · Nature Microbiology]

Structure of the complete extracellular bacterial flagellum reveals mechanism for flagellin incorporation

Corresponding Author: Dr Julien Bergeron

Version 0:

Reviewer comments:

Reviewer #1

(Remarks to the Author)

This is a very strong paper that is well suited for publication in Nature Microbiology. An atomic model for the complete extracellular flagellum is a significant advance, and will be of interest to many people who study bacterial motility. I had only minor comments that should be easily addressed by the authors.

Line 47) "the filament is a helical assembly". Actually, it is a superhelical assembly, and there is no mention of supercoiling anywhere in the paper (it does appear in the title of two references, however). Mutants have been studied for many years that have strictly helical (and therefore straight) flagellar filaments, and these mutants are all non-motile.

Line 86) "In this map, we observed a seam in the cap". Given the very poor resolution, this statement does not seem well supported.

Lines 93-104) It would be nice to characterize the motility (or lack of it) for these short filaments.

Line 114) "structural similarity between FlgK and FlgL". What is the percent identity?

Lines 113-114) 14 copies of FliC are included in the model. It appears that no helical symmetry has been imposed, so one could ask whether these are supercoiled or for some reason helical. The differences between the two over such a short region are subtle, but should be detectable. Such small differences are discussed in Kreuzberger et al., Cell (2022).

Line 431) "between left-handed and right-handed helical assemblies". The hand here is referring to the macroscopic supercoil, so this should be something like "between left-handed and right-handed supercoiled assemblies"

Lines 438-440) "the flexible hook can effectively transmit rotational force to the rigid filament without passing along the mechanical stress caused by its flexibility". I cannot understand this. How does the HFJ structure explain how mechanical stress is not transmitted?

Methods) It is absurd to use resolutions specified to a hundredth of an Å, as this precision has no significance given the large role of a mask in determining the stated resolution. It would help if more common sense existed in the field concerning this matter. For example, if one were using a digital thermometer, I doubt that anyone would write that the experiment was done at 4.03 °C. Common sense dictates that one writes 4 °C.

Reviewer #3

(Remarks to the Author)

The manuscript by Einkenkel, Qin et al. presents remarkable structural insights into the native, complete bacterial flagellum, with particular focus on the hook-filament junction and filament cap. Using single-particle cryo-EM, the authors have solved two critical structures from *Salmonella enterica*: the assembled hook-filament junction (HFJ) and the filament cap complex (FliD/FliC), both of which had previously eluded structural characterization.

Their methodological approach combines cryo-EM/ET with sophisticated conformational flexibility analysis (using CryoSPARC's approach), followed by structure-guided mutagenesis and functional assays. This comprehensive strategy enables the authors to propose a convincing mechanism for flagellin incorporation and filament assembly. Additionally, they solved a cryo-EM structure from a *Campylobacter jejuni* mutant strain that provides crucial insights into early assembly states, capturing the FliD cap complexed with the HFJ before flagellin incorporation.

The discovery of the asymmetric organization of the FliD pentamer capping the filament, along with the identification of a chaperone-like cavity for flagellin folding, represents a major advance in the field. The analysis of conformational dynamics using CryoSPARC's 3D Variability Analysis further enriches this understanding by revealing FliD/FliC conformational changes that illuminate the molecular mechanism of filament assembly.

This work represents a tour de force that provides groundbreaking evidence, significantly advancing our understanding of bacterial flagellar motility. The figures and movies are excellent, the narrative is clear, and the discussion effectively contextualizes these advances within the current state of the field. This paper will be of broad interest to the microbiology and structural biology communities.

Some minor corrections should be considered, and some sections could probably be improved to better address the limitations of the study.

Major comments.

1. Conformational Flexibility Analysis.

The continuous flexibility analysis approach used is fundamental for the section "Conformational changes to the cap upon filament assembly are coupled with flagellin folding" and consequently for the authors' mechanistic interpretation of the filament assembly process. While this cutting-edge approach is powerful for visualizing conformational heterogeneity at the single particle level, there are well-known risks regarding fitting models/maps to individual particles (i.e., noisy 2D projections of heterogeneous population volumes, with different CTF corruptions, etc). These can lead to overfitting, or even to artifactual deformations (for instance considering "bad particle" subpopulations).

Several aspects require attention:

a) The Methods section should detail specifics of how 3DVA was performed:

- Number of particles included in the analysis
- Number of variability components searched
- Were low/high-pass filters used?

b) A supplementary figure would be helpful to show:

- 2D scatter plots of latent coordinates of individual particles for the variability components (illustrating variability extent across different dimensions)
- UMAP embedding of 3DVA latent coordinates, or other analogous clustering approaches
- If clusters are identified, histograms of estimated contrast alpha scales per-particle, to quantify agreement between particles and consensus 3D density

c) The main conformational rearrangements should be validated using at least one alternative well-established approach beyond CryoSPARC's 3DVA (e.g., cryoDRGN, and/or others)

d) Movie S3, currently the only place showing different conformation maps, could be improved to better link it to Fig 3A,B:

- Provide as supplementary a sequence of labeled snapshots along the "movie", showing intermediates that are properly labeled and colored to match Fig 3

2. Structure-Function Relationships.

To better understand the evidence for Fig 3D, a clearer connection seems needed between analyzed structural residues and subsequent functional mutations. Extended Data Figure 3 is worth improving:

- Please add residue numbers, so that a clear connection to functional analyses presented in Figs 4 and 5 is ensured
- Current blue/yellow/green highlights to pinpoint interactions show inconsistencies (as far as I understand them): please correct

3. Hook-Filament Junction Mechanics.

The discussion proposes that a two-layered junction is required to prevent direct FlgE-FliC interaction, ensuring proper uncoupling between flexible and rigid rotating polymers (lines 437-441). However, several mechanical aspects need clarification:

- The mechanical nature of FlgL and FlgK themselves (flexible vs. rigid)
- It seems that the fundamental question here still persists: how is the FlgE polymer (with protomers expanding and contracting through the rotation cycle such that a defined polymer angle is preserved) linked to the FlgK layer? Please elaborate further.
- Please also elaborate further about potential hook heterogeneity in both *S. enterica* and *C. jejuni* structures: especially in *S. enterica* the hook density seems to be weak...
- Please clarify the statement "density of the FlgKL junction is mixed with the density of FlgE hook and FliC filament" (lines 632-633): what do you mean by "mixing"?
- Could you perform a 3DVA analysis of these particles to better understand how flexibility transitions from the FlgE to all the way to the FliC assemblies? (discretely? continuously?)

Minor Comments

1. Map Refinement and Resolution Assessment

- Lines 107-108: The statement "...we were able to refine this map further, to 3.3 Å..." creates confusion. The chronological sequence appears to be: initial 3.3 Å map showing discrete heterogeneity, followed by 3D classification and refinement leading to the final 3.7 Å reconstruction used for main analyses. This should be rephrased for clarity.

• Quality Assessment and Refinement:

- Q-scores appear lower than expected for the nominal FSC resolutions in both cap:filament and HFJ structures, with this discrepancy being more pronounced in the latter
- Consider re-refinement in reciprocal space using Servalcat (Yamashita et al. 2021 Acta Cryst D 77:1282), fully exploiting advances in protein restraints definitions (Yamashita et al. 2023 Acta Cryst D 79:368).
- Explanation needed for refining the HFJ model at lower resolution than the nominal SGFSC reconstruction resolution - are there signs of resolution overestimation?

2. Data Presentation and Technical Details

- Line 145: Text indicates a different measurement for the short axis of the oval cavity size compared to Fig 2A. Should be ~36Å to match the figure (or figure needs correction)
- Mutagenesis Strategy:
 - Rationale should be provided for using only serines or arginines as substituent amino acids
 - Several mutations show discrepancies between text and figures, these inconsistencies should be resolved: F461S and L450S appear only in text ; Y456S shown only in Fig 4C.

3. Figure Corrections and Species Comparisons

- Lines 277 and 374: Figure references appear to be incorrect, likely should refer to Fig 1 rather than Fig 2?
- Fig 6G-H require additional clarification:
 - Please elaborate further about the significant difference in FliD heads (D1-D2 angle) between *S. enterica* and *C. jejuni* caps (56° vs 106°)
 - Are there potential impacts on the pentamer structures due to this?
 - It is not clear to me whether this is related to your paragraph lines 387-395?
 - The D1-D2 hinge angle disparity should be visible in Fig 6H's superposition: could you label this better to understand what is happening? I agree with the authors in that these differences may correspond to species-specific traits, yet they are indeed important features to be shown.

Decision Letter:

9th December 2024

Dear Dr Bergeron,

Thank you for your patience while your manuscript "Structure of the complete extracellular bacterial flagellum reveals mechanism for flagellin incorporation" was under peer-review at Nature Microbiology. It has now been seen by 2 referees, whose expertise and comments you will find at the end of this email. Although they find your work of interest, they have raised several concerns that will need to be addressed before we can consider publication of the work in Nature Microbiology.

In particular, Referee #2 asks that the conformational flexibility analysis be further validated, and also requests more information on the spread and variability of the analysis plus more methodological details to rule out the possibility of overinterpretation that they note can arise with such approaches. They also ask for more insight into the hook-filament junction mechanics including whether it would be possible to apply 3DVA here. In addition to this, Referee #1 requests that the motility of short filament mutants be characterised. We feel that these are critical points which would need to be addressed for us to further consider a revised manuscript, alongside the remaining issues outlined in the referees' reports, which are clear and should be straightforward to address.

Should further experimental data allow you to address these criticisms, we would be happy to look at a revised manuscript.

Please include a data availability statement as a separate section after Methods but before references, under the heading "Data Availability". This section should inform readers about the availability of the data used to support the conclusions of your study. This information includes accession codes to public repositories (data banks for protein, DNA or RNA sequences, microarray, proteomics data etc...), references to source data published alongside the paper, unique identifiers such as URLs to data repository entries, or data set DOIs, and any other statement about data availability. At a minimum, you should include the following statement: "The data that support the findings of this study are available from the corresponding author upon request", mentioning any restrictions on availability. If DOIs are provided, we also strongly encourage including these in the Reference list (authors, title, publisher (repository name), identifier, year). For more guidance on how to write this section please see: <http://www.nature.com/authors/policies/data/data-availability-statements-data-citations.pdf>

* If you have not done so already we suggest that you begin to revise your manuscript so that it conforms to our Article format instructions at <http://www.nature.com/nmicrobiol/info/final-submission>. Refer also to any guidelines provided in this letter.

When submitting the revised version of your manuscript, please pay close attention to our [href="https://www.nature.com/nature-portfolio/editorial-policies/image-integrity">Digital Image Integrity Guidelines.](https://www.nature.com/nature-portfolio/editorial-policies/image-integrity) and to the following points below:

Link Redacted

Note: This url links to your confidential homepage and associated information about manuscripts you may have submitted or be reviewing for us. If you wish to forward this e-mail to co-authors, please delete this link to your homepage first.

Nature Microbiology is committed to improving transparency in authorship. As part of our efforts in this direction, we are now requesting that all authors identified as 'corresponding author' on published papers create and link their Open Researcher and Contributor Identifier (ORCID) with their account on the Manuscript Tracking System (MTS), prior to acceptance. This applies to primary research papers only. ORCID helps the scientific community achieve unambiguous attribution of all scholarly contributions. You can create and link your ORCID from the home page of the MTS by clicking on 'Modify my Springer Nature account'. For more information please visit [please visit www.springernature.com/orcid](http://www.springernature.com/orcid).

If you wish to submit a suitably revised manuscript we would hope to receive it within 6 months. If you cannot send it within this time, please let us know. We will be happy to consider your revision, even if a similar study has been accepted for publication at Nature Microbiology or published elsewhere (up to a maximum of 6 months).

Yours sincerely,

Reviewer Expertise:

Referee #1: cyroEM bacterial surface structures
Referee #2: flagella, motility, structural analyses

Reviewer Comments:

Reviewer #1 (Remarks to the Author):

This is a very strong paper that is well suited for publication in Nature Microbiology. An atomic model for the complete extracellular flagellum is a significant advance, and will be of interest to many people who study bacterial motility. I had only minor comments that should be easily addressed by the authors.

Line 47) "the filament is a helical assembly". Actually, it is a superhelical assembly, and there is no mention of supercoiling anywhere in the paper (it does appear in the title of two references, however). Mutants have been studied for many years that have strictly helical (and therefore straight) flagellar filaments, and these mutants are all non-motile.

Line 86) "In this map, we observed a seam in the cap". Given the very poor resolution, this statement does not seem well supported.

Lines 93-104) It would be nice to characterize the motility (or lack of it) for these short filaments.

Line 114) “structural similarity between FlgK and FlgL”. What is the percent identity?

Lines 113-114) 14 copies of FliC are included in the model. It appears that no helical symmetry has been imposed, so one could ask whether these are supercoiled or for some reason helical. The differences between the two over such a short region are subtle, but should be detectable. Such small differences are discussed in Kreutzberger et al., Cell (2022).

Line 431) “between left-handed and right-handed helical assemblies”. The hand here is referring to the macroscopic supercoil, so this should be something like “between left-handed and right-handed supercoiled assemblies”

Lines 438-440) “the flexible hook can effectively transmit rotational force to the rigid filament without passing along the mechanical stress caused by its flexibility”. I cannot understand this. How does the HFJ structure explain how mechanical stress is not transmitted?

Methods) It is absurd to use resolutions specified to a hundredth of an Å, as this precision has no significance given the large role of a mask in determining the stated resolution. It would help if more common sense existed in the field concerning this matter. For example, if one were using a digital thermometer, I doubt that anyone would write that the experiment was done at 4.03 °C. Common sense dictates that one writes 4 °C.

Reviewer #2 (Remarks to the Author):

The manuscript by Eickenel, Qin et al. presents remarkable structural insights into the native, complete bacterial flagellum, with particular focus on the hook-filament junction and filament cap. Using single-particle cryo-EM, the authors have solved two critical structures from *Salmonella enterica*: the assembled hook-filament junction (HFJ) and the filament cap complex (FliD/FliC), both of which had previously eluded structural characterization.

Their methodological approach combines cryo-EM/ET with sophisticated conformational flexibility analysis (using CryoSPARC's approach), followed by structure-guided mutagenesis and functional assays. This comprehensive strategy enables the authors to propose a convincing mechanism for flagellin incorporation and filament assembly. Additionally, they solved a cryo-EM structure from a *Campylobacter jejuni* mutant strain that provides crucial insights into early assembly states, capturing the FliD cap complexed with the HFJ before flagellin incorporation.

The discovery of the asymmetric organization of the FliD pentamer capping the filament, along with the identification of a chaperone-like cavity for flagellin folding, represents a major advance in the field. The analysis of conformational dynamics using CryoSPARC's 3D Variability Analysis further enriches this understanding by revealing FliD/FliC conformational changes that illuminate the molecular mechanism of filament assembly.

This work represents a tour de force that provides groundbreaking evidence, significantly advancing our understanding of bacterial flagellar motility. The figures and movies are excellent, the narrative is clear, and the discussion effectively contextualizes these advances within the current state of the field. This paper will be of broad interest to the microbiology and structural biology communities.

Some minor corrections should be considered, and some sections could probably be improved to better address the limitations of the study.

Major comments.

1. Conformational Flexibility Analysis.

The continuous flexibility analysis approach used is fundamental for the section “Conformational changes to the cap upon filament assembly are coupled with flagellin folding” and consequently for the authors’ mechanistic interpretation of the filament assembly process. While this cutting-edge approach is powerful for visualizing conformational heterogeneity at the single particle level, there are well-known risks regarding fitting models/maps to individual particles (i.e., noisy 2D projections of heterogeneous population volumes, with different CTF corrections, etc). These can lead to overfitting, or even to artifactual deformations (for instance considering “bad particle” subpopulations).

Several aspects require attention:

a) The Methods section should detail specifics of how 3DVA was performed:

- Number of particles included in the analysis
- Number of variability components searched
- Were low/high-pass filters used?

b) A supplementary figure would be helpful to show:

- 2D scatter plots of latent coordinates of individual particles for the variability components (illustrating variability extent across different dimensions)
- UMAP embedding of 3DVA latent coordinates, or other analogous clustering approaches
- If clusters are identified, histograms of estimated contrast alpha scales per-particle, to quantify agreement between particles and consensus 3D density

c) The main conformational rearrangements should be validated using at least one alternative well-established approach beyond CryoSPARC's 3DVA (e.g., cryoDRGN, and/or others)

d) Movie S3, currently the only place showing different conformation maps, could be improved to better link it to Fig 3A,B:

- Provide as supplementary a sequence of labeled snapshots along the "movie", showing intermediates that are properly labeled and colored to match Fig 3

2. Structure-Function Relationships.

To better understand the evidence for Fig 3D, a clearer connection seems needed between analyzed structural residues and subsequent functional mutations. Extended Data Figure 3 is worth improving:

- Please add residue numbers, so that a clear connection to functional analyses presented in Figs 4 and 5 is ensured
- Current blue/yellow/green highlights to pinpoint interactions show inconsistencies (as far as I understand them): please correct

3. Hook-Filament Junction Mechanics.

The discussion proposes that a two-layered junction is required to prevent direct FlgE-FlhC interaction, ensuring proper uncoupling between flexible and rigid rotating polymers (lines 437-441). However, several mechanical aspects need clarification:

- The mechanical nature of FlgL and FlgK themselves (flexible vs. rigid)
- It seems that the fundamental question here still persists: how is the FlgE polymer (with protomers expanding and contracting through the rotation cycle such that a defined polymer angle is preserved) linked to the FlgK layer? Please elaborate further.
- Please also elaborate further about potential hook heterogeneity in both *S. enterica* and *C. jejuni* structures: especially in *S. enterica* the hook density seems to be weak...
- Please clarify the statement "density of the FlgKL junction is mixed with the density of FlgE hook and FlhC filament" (lines 632-633): what do you mean by "mixing"?
- Could you perform a 3DVA analysis of these particles to better understand how flexibility transitions from the FlgE to all the way to the FlhC assemblies? (discretely? continuously?)

Minor Comments

1. Map Refinement and Resolution Assessment

- Lines 107-108: The statement "...we were able to refine this map further, to 3.3 Å..." creates confusion. The chronological sequence appears to be: initial 3.3 Å map showing discrete heterogeneity, followed by 3D classification and refinement leading to the final 3.7 Å reconstruction used for main analyses. This should be rephrased for clarity.
- Quality Assessment and Refinement:
 - Q-scores appear lower than expected for the nominal FSC resolutions in both cap:filament and HFJ structures, with this discrepancy being more pronounced in the latter
 - Consider re-refinement in reciprocal space using Servalcat (Yamashita et al. 2021 Acta Cryst D 77:1282), fully exploiting advances in protein restraints definitions (Yamashita et al. 2023 Acta Cryst D 79:368).
 - Explanation needed for refining the HFJ model at lower resolution than the nominal SGFSC reconstruction resolution - are there signs of resolution overestimation?

2. Data Presentation and Technical Details

- Line 145: Text indicates a different measurement for the short axis of the oval cavity size compared to Fig 2A. Should be ~36Å to match the figure (or figure needs correction)
- Mutagenesis Strategy:
 - Rationale should be provided for using only serines or arginines as substituent amino acids
 - Several mutations show discrepancies between text and figures, these inconsistencies should be resolved: F461S and L450S appear only in text ; Y456S shown only in Fig 4C.

3. Figure Corrections and Species Comparisons

- Lines 277 and 374: Figure references appear to be incorrect, likely should refer to Fig 1 rather than Fig 2?
- Fig 6G-H require additional clarification:
 - Please elaborate further about the significant difference in FlhD heads (D1-D2 angle) between *S. enterica* and *C. jejuni* caps (56° vs 106°)
 - Are there potential impacts on the pentamer structures due to this?
 - It is not clear to me whether this is related to your paragraph lines 387-395?
 - The D1-D2 hinge angle disparity should be visible in Fig 6H's superposition: could you label this better to understand what is happening? I agree with the authors in that these differences may correspond to species-specific traits, yet they are indeed important features to be shown.

Version 1:

Reviewer comments:

Reviewer #1

(Remarks to the Author)

The authors have done an excellent job of addressing all of my concerns.

Reviewer #3

(Remarks to the Author)

I want to thank the authors for having responded to all the comments I had brought up, and have done it very satisfactorily.

A very small excerpt here: the program Servalcat (contrary to what could be thought, I believe legitimately so, since it was indeed used for several (ultra)high resolution cryoEM structure refinements), can be used VERY efficiently with medium and even low resolution cryoEM maps (I have had excellent results even at 5Å e.g.). In such low-res cases of course internal symmetry is a good restraint to be used if present (--ncsr option), but also super powerful are initial self-restraints (using well refined startign models, including AlphaFold or Rx structures), which are very well managed with the --jellybody approach, among other tools.

Decision Letter:

Our ref: NMICROBIOL-24103161A

25th March 2025

Dear Dr. Bergeron,

Thank you for submitting your revised manuscript "Structure of the complete extracellular bacterial flagellum reveals mechanism for flagellin incorporation" (NMICROBIOL-24103161A). It has now been seen by the original referees and their comments are below. The reviewers find that the paper has improved in revision, and therefore we'll be happy in principle to publish it in Nature Microbiology, pending minor revisions to satisfy the referees' final requests and to comply with our editorial and formatting guidelines.

Thank you again for your interest in Nature Microbiology Please do not hesitate to contact me if you have any questions.

Sincerely,

Reviewer #1 (Remarks to the Author):

The authors have done an excellent job of addressing all of my concerns.

Reviewer #3 (Remarks to the Author):

I want to thank the authors for having responded to all the comments I had brought up, and have done it very satisfactorily.

A very small excerpt here: the program Servalcat (contrary to what could be thought, I believe legitimately so, since it was indeed used for several (ultra)high resolution cryoEM structure refinements), can be used VERY efficiently with medium and even low resolution cryoEM maps (I have had excellent results even at 5Å e.g.). In such low-res cases of course internal symmetry is a good restraint to be used if present (--ncsr option), but also super powerful are initial self-restraints (using well refined startign models, including AlphaFold or Rx structures), which are very well managed with the --jellybody approach, among other tools.

Version 2:

Decision Letter:

19th May 2025

Dear Dr Bergeron,

I am pleased to accept your Article "Structure of the complete extracellular bacterial flagellum reveals mechanism for flagellin incorporation" for publication in Nature Microbiology. Thank you for having chosen to submit your work to us and many congratulations.

Authors may need to take specific actions to achieve [compliance](https://www.springernature.com/gp/open-research/funding/policy-compliance-faqs) with funder and institutional open access mandates. If your research is supported by a funder that requires immediate open access (e.g. according to [Plan S principles](https://www.springernature.com/gp/open-research/plan-s-compliance)) then you should select the gold OA route, and we will direct you to the compliant route where possible. For authors selecting the subscription publication route, the journal's standard licensing terms will need to be accepted, including [self-archiving policies](https://www.nature.com/nature-portfolio/editorial-policies/self-archiving-and-license-to-publish). Those licensing terms will supersede any other terms that the author or any third party may assert apply to any version of the manuscript.

With kind regards,

P.S. Click on the following link if you would like to recommend Nature Microbiology to your librarian
<http://www.nature.com/subscriptions/recommend.html#forms>

** Visit the Springer Nature Editorial and Publishing website at www.springernature.com/editorial-and-publishing-jobs for more information about our career opportunities. If you have any questions please click here.**

Reviewer Expertise:

Referee #1: cryoEM bacterial surface structures

Referee #2: flagella, motility, structural analyses

Reviewer Comments:

Reviewer #1 (Remarks to the Author):

This is a very strong paper that is well suited for publication in Nature Microbiology. An atomic model for the complete extracellular flagellum is a significant advance, and will be of interest to many people who study bacterial motility. I had only minor comments that should be easily addressed by the authors.

RE: Thank you very much for your positive feedback and appreciation of our work. We are pleased to hear that you recognize the significance of our findings, and we have carefully addressed your comments below.

Line 47) “the filament is a helical assembly”. Actually, it is a superhelical assembly, and there is no mention of supercoiling anywhere in the paper (it does appear in the title of two references, however). Mutants have been studied for many years that have strictly helical (and therefore straight) flagellar filaments, and these mutants are all non-motile.

RE: Thank you for this important clarification. We agree with the correction and have revised the text to refer to the filament as a superhelical assembly (**Page 2, Line 47**). We further performed 3D classification to analyze the filament’s conformation and found it to exhibit structural heterogeneity, with a predominantly supercoiled arrangement (Details of the analysis are provided below in the response to the comment to Lines 113-114 and in the **Response Figure 1**). We have also included a relevant reference to further support this statement (Kreutzberger et al., 2022, Cell).

Line 86) “In this map, we observed a seam in the cap”. Given the very poor resolution, this statement does not seem well supported.

RE: Thank you for pointing this out. We noticed the ‘soft’ lighting mode of the map we used before didn’t show the seam clearly. The lighting has been changed to show the seam on the FliD cap and the tilted plane clearly. The figures have been updated in the revised manuscript (**Extended Data Figure 1b,c**).

Lines 93-104) It would be nice to characterize the motility (or lack of it) for these short filaments.

RE: Thank you for this insightful suggestion. To address this point, we analyzed the single-cell swimming velocities of the hyperflagellated short filament mutant under three conditions: without fliC induction, after 30 minutes of induction, and after 120 minutes of induction, and compared them to a hyperflagellated "wild-type" strain (**Extended Data Figure 1e**). We also quantified the filament length of the mutant after 120 minutes of fliC induction (**Extended Data Figure 1f**). As previously reported (Halte

et al., 2024 (10.1101/2024.06.28.599820)), Salmonella requires a minimum filament length of ~2.5 μm to achieve motility. Consistent with this, our analysis confirmed that the short filaments produced by the mutant did not reach this critical length and, as a result, the mutant was non-motile at the induction time used for hook-basal body purification. This has been indicated in the text (**Page 4, Lines 102-105**).

Line 114) “structural similarity between FlgK and FlgL”. What is the percent identity?

RE: Thank you for raising this point. To clarify, a pairwise structural alignment of FlgK and FlgL using TM-align (RCSB PDB), indicated a 14% sequence identity between these two proteins. Despite the low sequence identity, the structural alignment indicates moderate similarity in the overall fold of these two proteins, with a RMSD of 4.7 \AA , which is consistent with what is commonly observed for flagellar axial proteins. We changed the manuscript accordingly (**Page 4, Lines 117-119**).

Lines 113-114) 14 copies of FliC are included in the model. It appears that no helical symmetry has been imposed, so one could ask whether these are supercoiled or for some reason helical. The differences between the two over such a short region are subtle, but should be detectable. Such small differences are discussed in Kreutzberger et al., Cell (2022).

RE: Thank you for raising this important point regarding supercoiling. While our filament structure was processed without applying symmetry, whether it adopted any curvature was not immediately apparent. However, for this analysis, we used a smaller box size (540x540 \AA) compared to the study of Kreutzberger et al. (674x674 \AA). They reported that using a smaller box size resulted in straight volumes with minimal curvature due to the refinement process. This difference in box size may explain why the curvature of our filament is less apparent.

To address this, we performed 3D classification to separate filament conformations and refined their respective maps (See **Response Figure 1** below). This led to two maps, with distinct conformations. In the first map (344,897 particles), the filament is straight and adopts a helical symmetry. In the 2nd map (299,043 particles), we observed a slight bent in the filament structure, corresponding to a curved conformation (as indicated by arrows). This suggests that in our purified complex, the filament is heterogeneous in structure, and largely supercoiled, but with a population of particles that adopt helical symmetry. We did not include this data in the revised manuscript, as it is not central to our message, and the relevance of this observation is not known as we were using short filaments for our cryo-EM studies.

Response Figure 1. Structural Comparison of filament conformations.

a, 3D classes of a straight (left) and curved (right) filament. **b**, Cross-sectional views highlighting structural differences between straight and curved filaments.

Line 431) “between left-handed and right-handed helical assemblies”. The hand here is referring to the macroscopic supercoil, so this should be something like “between left-handed and right-handed supercoiled assemblies”

RE: We agree with this correction and have updated the text accordingly to clarify that the filament alternates between left-handed and right-handed supercoiled assemblies in response to changes in motor rotation (**Page 23, Line 472**).

Lines 438-440) “the flexible hook can effectively transmit rotational force to the rigid filament without passing along the mechanical stress caused by its flexibility”. I cannot understand this. How does the HFJ structure explain how mechanical stress is not transmitted?

RE: Thank you for pointing this out, we agree that this aspect required further clarification in the manuscript. Due to the bending of the hook and HFJ, its protofilaments can be structurally divided into compressed and extended states, as previously shown for the hook (Kato et al., 2019 (<https://doi.org/10.1038/s41467-019->

13252-9), Samatey et al., 2004 (<https://doi.org/10.1038/nature02997>)). By comparing these regions and analysing the flexibility transitions of the HFJ with 3DVA, we observed that FlgK may play a role in relieving mechanical stress from the hook due to its elasticity. To clarify these findings and their implications, we have revised the manuscript (**Page 16, Lines 320-340**). We also added **panel b to Figure 5**, which illustrates the FlgE-FlgK-FlgL-FliC protofilament in both its extended and compressed states. Additionally, we included **Extended Data Fig. 7a-d**, showing superimpositions of the extended and compressed states of the protofilament components and **Movie S5** and **Extended Data Fig. 7g-j**, which shows the 3DVA of the HFJ as supporting evidence.

Methods) It is absurd to use resolutions specified to a hundredth of an Å, as this precision has no significance given the large role of a mask in determining the stated resolution. It would help if more common sense existed in the field concerning this matter. For example, if one were using a digital thermometer, I doubt that anyone would write that the experiment was done at 4.03 °C. Common sense dictates that one writes 4 °C.

RE: We appreciate the reviewer's comment regarding the precision of reported resolutions. We have rounded the reported resolutions to one decimal place throughout the manuscript (e.g., 3.74 Å → 3.7 Å) to reflect a more realistic level of precision.

Reviewer #2 (Remarks to the Author):

The manuscript by Einkenel, Qin et al. presents remarkable structural insights into the native, complete bacterial flagellum, with particular focus on the hook-filament junction and filament cap. Using single-particle cryo-EM, the authors have solved two critical structures from *Salmonella enterica*: the assembled hook-filament junction (HFJ) and the filament cap complex (FliD/FliC), both of which had previously eluded structural characterization.

Their methodological approach combines cryo-EM/ET with sophisticated conformational flexibility analysis (using CryoSPARC's approach), followed by structure-guided mutagenesis and functional assays. This comprehensive strategy enables the authors to propose a convincing mechanism for flagellin incorporation and filament assembly. Additionally, they solved a cryo-EM structure from a *Campylobacter jejuni* mutant strain that provides crucial insights into early assembly states, capturing the FliD cap complexed with the HFJ before flagellin incorporation.

The discovery of the asymmetric organization of the FliD pentamer capping the filament, along with the identification of a chaperone-like cavity for flagellin folding, represents a major advance in the field. The analysis of conformational dynamics using CryoSPARC's 3D Variability Analysis further enriches this understanding by revealing FliD/FliC conformational changes that illuminate the molecular mechanism of filament assembly.

This work represents a tour de force that provides groundbreaking evidence, significantly advancing our understanding of bacterial flagellar motility. The figures and

movies are excellent, the narrative is clear, and the discussion effectively contextualizes these advances within the current state of the field. This paper will be of broad interest to the microbiology and structural biology communities.

Some minor corrections should be considered, and some sections could probably be improved to better address the limitations of the study.

RE: Thank you very much for your thorough and thoughtful review of our manuscript, and for recognizing the significance of our structural and functional findings. We have carefully considered your suggestions regarding corrections and possible improvements to specific sections, specifically for the validation of the flexibility analysis and the analysis of the HFJ using 3DVA. Below, we provide detailed responses to each point, outlining the steps we have taken to address your comments and improve the manuscript accordingly.

Major comments.

1. Conformational Flexibility Analysis.

The continuous flexibility analysis approach used is fundamental for the section “Conformational changes to the cap upon filament assembly are coupled with flagellin folding” and consequently for the authors’ mechanistic interpretation of the filament assembly process. While this cutting-edge approach is powerful for visualizing conformational heterogeneity at the single particle level, there are well-known risks regarding fitting models/maps to individual particles (i.e., noisy 2D projections of heterogeneous population volumes, with different CTF corruptions, etc). These can lead to overfitting, or even to artifactual deformations (for instance considering “bad particle” subpopulations).

Several aspects require attention:

a) The Methods section should detail specifics of how 3DVA was performed:

- Number of particles included in the analysis
- Number of variability components searched
- Were low/high-pass filters used?

RE: Thank you for your request regarding the details of the 3DVA analysis. We have included this information in the Methods section (**Pages 31-32, Lines 683-688**), specifying that the analysis involved 42,342 particles, three variability components, an 8Å filter, and a mask covering the entire FliD pentamer along with its adjacent FliC subunits. No high-pass filter was applied.

b) A supplementary figure would be helpful to show:

- 2D scatter plots of latent coordinates of individual particles for the variability components (illustrating variability extent across different dimensions)
- UMAP embedding of 3DVA latent coordinates, or other analogous clustering approaches
- If clusters are identified, histograms of estimated contrast alpha scales per-particle,

to quantify agreement between particles and consensus 3D density

RE: Thank you for your suggestions. Plots of 3DVA and clusters have been included in the **Extended Data Figure 3** (filament cap) and **Extended Data Figure 7** (HFJ).

c) The main conformational rearrangements should be validated using at least one alternative well-established approach beyond CryoSPARC's 3DVA (e.g., cryoDRGN, and/or others)

RE: Thank you for your valuable suggestion. We agree that validating the main conformational rearrangements is important.

To ensure the robustness of our conclusions from the 3DVA analysis, we performed an independent validation using non-uniform refinement on particle stacks derived from the 3DVA clusters (See **Extended Data Figure 3** in the revised manuscript). By leveraging the same dataset and refining distinct particle subsets, this approach directly validates the observed conformational states.

Specifically, we reconstructed intermediate states using the particles in Clusters 12, 4, 15, 1, which contained the highest number of particles. While these maps could not be refined to high resolution due to lower particle number, their density contours remain well-defined, allowing us to clearly distinguish protein features. Importantly, these maps unambiguously confirm that the conformational intermediates observed in the 3DVA analysis accurately reflect the states present in the cryo-EM data (**Extended Data Figure 3g**).

We have incorporated the validation of the 3DVA into the manuscript (**Extended Data Figure 3** and **Page 9, Lines 194-201**).

d) Movie S3, currently the only place showing different conformation maps, could be improved to better link it to Fig 3A,B:

- Provide as supplementary a sequence of labeled snapshots along the “movie”, showing intermediates that are properly labeled and colored to match Fig 3

RE: Thank you for your helpful suggestion. To better link Movie S3 with Figure 3a,b, we have updated the movie by colouring the densities of FliDs and the incorporating FliC to match the corresponding elements in the figure (**Movie S3, Movie S4**). Since Figure 3 is already composed of snapshots from this movie, we feel that an additional sequence of labelled snapshots would be somewhat redundant. We hope that the updated colouring scheme provides a clearer visual connection between the movie and the figure.

2. Structure-Function Relationships.

To better understand the evidence for Fig 3D, a clearer connection seems needed between analyzed structural residues and subsequent functional mutations.

RE: Thank you for this valuable suggestion. To better clarify the structure-function relationship, we have revised the first paragraph of the section “FliD terminal regions mediate flagellin insertion at the flagellar distal end” to more explicitly link the structural observations to the subsequent mutagenesis and functional assays (**Page 9, Lines 217-222**). Further, we updated the last paragraph of the section to clarify that the observed

motility and filament assembly defects in the mutants align with the structural evidence supporting the proposed model in Figure 3d (Page 13, Lines 296-299).

To further improve clarity, we have updated **Figure 3d** to highlight the involvement of key residues, F440 and V9, which are central to the FliD-FliC interaction interface and that were most characterized. These residues were specifically chosen for mutational analysis based on our structural model, which revealed their roles in stabilizing filament elongation through interactions between FliD D0 and flagellin.

Extended Data Figure 3 is worth improving:

- Please add residue numbers, so that a clear connection to functional analyses presented in Figs 4 and 5 is ensured
- Current blue/yellow/green highlights to pinpoint interactions show inconsistencies (as far as I understand them): please correct

RE: Thank you for your feedback on Extended Data Figure 3 (now **Extended Data Figure 4**). We agree that adding residue numbers will improve clarity and better connect this figure to the functional analyses in Figures 4 and 5. We have now incorporated residue numbers to explicitly indicate key interactions and ensure consistency with the functional data. We also deleted the highlights to show the interaction interfaces to avoid confusion (now: **Extended Data Figure 4**).

3. Hook-Filament Junction Mechanics.

The discussion proposes that a two-layered junction is required to prevent direct FlgE-FliC interaction, ensuring proper uncoupling between flexible and rigid rotating polymers (lines 437-441). However, several mechanical aspects need clarification:

- The mechanical nature of FlgL and FlgK themselves (flexible vs. rigid)
- It seems that the fundamental question here still persists: how is the FlgE polymer (with protomers expanding and contracting through the rotation cycle such that a defined polymer angle is preserved) linked to the FlgK layer? Please elaborate further.
- Could you perform a 3DVA analysis of these particles to better understand how flexibility transitions from the FlgE to all the way to the FliC assemblies? (discretely? continuously?)

RE: Thank you for these insightful comments. We agree that further clarification was needed regarding the mechanical nature of the HFJ and its role as a connecting platform between the hook and the filament.

To address these points, we performed 3DVA analysis as suggested to visualise conformational changes during the flexibility transitions from FlgE in the hook through the HFJ and finally to the filament. Additionally, we analysed the structural differences between the extended and compressed states of each protofilament component.

Our findings indicate that FlgK undergoes significant structural shifts, with its D0 domain and the uppermost part of the D2 domain bending toward the lumen in the compressed state. In contrast, FlgL exhibits a more moderate shift, with only the D0 domain rising and bending slightly inward, while FliC remains largely unchanged.

Based on these results, we propose that the FlgKL junction functions as a mechanical buffer that absorbs stress from the flexible hook. Specifically, we suggest that the elasticity of FlgK plays a key role in preventing mechanical stress from propagating to the flagellin filament. Additionally, our model suggests that FlgK's D1b and D2 domains interact with the D1 and D2 domains of FlgE, potentially limiting FlgE deformation and therefore stabilizing the FlgE-FlgK interface. However, due to the resolution constraints in these regions, we cannot resolve their molecular interactions in detail.

We have updated the manuscript (**Page 16, Lines 320-340**) to include these important elements of analysis. We have also expanded **Figure 5** by adding **panel b**, which depicts the FlgE-FlgK-FlgL-FliC protofilament in both its extended and compressed states. Additionally, we incorporated **Extended Data Fig. 7**, which presents superimposed views of the protofilament components in both states as well as the modeled combinations of the protofilament components. Furthermore, **Movie S5** and **Extended Data Fig. 7 g-j** shows the 3DVA of the HFJ as supporting evidence.

- Please also elaborate further about potential hook heterogeneity in both *S. enterica* and *C. jejuni* structures: especially in *S. enterica* the hook density seems to be weak...

RE: The *C. jejuni* hook is proposed to be more rigid than the *S. enterica* hook (Matsunami et al., 2016 ([10.1038/ncomms13425](https://doi.org/10.1038/ncomms13425))). In *Campylobacter* and in related bacteria, hook protein FlgE was found to possess an extra domain D4, enabling the hook to be more stable and robust ([10.1038/ncomms13425](https://doi.org/10.1038/ncomms13425)). It might be two different strategies for bacteria to achieve motility to adapt to the environment. The hook of *S. enterica* is known to acquire sufficient flexibility to bundle peritrichous flagella (Hiraoka et al., 2017 ([10.1038/srep46723](https://doi.org/10.1038/srep46723))) whereas *C. jejuni* hook does not require this. *C. jejuni* hook may be more rigid/stiff than the *S. enterica* hook in order to tolerate and/or transmit a greater amount of torque (Yoon et al., 2016 ([10.1038/srep35552](https://doi.org/10.1038/srep35552)); Henderson et al., 2020 ([10.1128/mbio.02286-19](https://doi.org/10.1128/mbio.02286-19))). The weak density of the hook in our structure might be due to the tight mask on FlgKL that didn't include much FlgE. We implemented these clarifications in the manuscript (**Page 23, Lines 484-490**).

- Please clarify the statement "density of the FlgKL junction is mixed with the density of FlgE hook and FliC filament" (lines 632-633): what do you mean by "mixing"?

RE: Thank you for pointing out the need for clarification. We have rephrased this section in the manuscript accordingly (**Page 32, Lines 703-710**). In our initial map, we observed ambiguous densities in the outer domains of FlgKL that likely originated from misalignment of particles. For example, the density adjacent to FliC D2 exhibited features resembling the FlgL D2 loop, even though this feature was absent in other FliC subunits. This suggested that particle misalignment during refinement resulted in overlapping density contributions from adjacent subunits. To resolve this, we performed 3D classification, which successfully separated two distinct junction conformations, leading to improved density maps.

Minor Comments

1. Map Refinement and Resolution Assessment

- Lines 107-108: The statement "...we were able to refine this map further, to 3.3 Å..." creates confusion. The chronological sequence appears to be: initial 3.3 Å map showing discrete heterogeneity, followed by 3D classification and refinement leading to the final 3.7 Å reconstruction used for main analyses. This should be rephrased for clarity.

RE: Thank you for pointing out this potential source of confusion. We agree that the description of the map refinement process needed clarification, and we have rephrased this section to better reflect the chronological sequence (**Page 4, Lines 110-114**).

- Quality Assessment and Refinement:

- Q-scores appear lower than expected for the nominal FSC resolutions in both cap:filament and HFJ structures, with this discrepancy being more pronounced in the latter

RE: The average Q-scores of the *Salmonella* filament-cap and HFJ models are approximately 0.4. This is in line with the expected Q-score values at the corresponding resolutions (See Pintile et al., Nat. Methods, 2020 (10.1038/s41592-020-0731-1)), although on the lower end. Local resolution estimations (see Extended Data Figure 2) show that the resolution of D2-D3 of each subunit is lower than their D0-D1. This is the cause of the somewhat low Q-score, which is negative in those regions. We do note a significantly lower Q-score for the *Campylobacter* HFJ-cap structure (average 0.2), which is to be expected considering the lower resolution. We added a comment on these values in the Methods section (*Salmonella*: **Page 32, Lines 716-719**; *Campylobacter*: **Page: 41, Lines: 989-990**).

- Consider re-refinement in reciprocal space using Servalcat (Yamashita et al. 2021 Acta Cryst D 77:1282), fully exploiting advances in protein restraints definitions (Yamashita et al. 2023 Acta Cryst D 79:368).

RE: We agree that Servalcat is indeed a powerful tool for model refinement. However, it is particularly useful for modeling ligands, and/or hydrogen atoms, particularly in very high-resolution maps. Considering the size of our atomic models, the relatively limited resolutions of our maps, and the high range of local resolutions across the maps, this is unlikely to provide significantly improved statistics for our models, which are refined to appropriate levels.

- Explanation needed for refining the HFJ model at lower resolution than the nominal SGFSC reconstruction resolution - are there signs of resolution overestimation?

RE: Thank you for pointing this out. We noticed the D2 domains of FlgK were not well solved, so we applied a local refinement with a tight mask on FlgKL junction and orientation search. According to the FSC plot in Extended Data Figure 2, there is no sign of resolution overestimation. We did not include the mask used for refinement when we deposited our structures in wwPDB, which explains the lower resolution for the map in the EMDV validation report. However, we note that this correlates with the 'No mask' resolution in the GSFSC plot of CryoSPARC (Extended Data Fig. 2).

2. Data Presentation and Technical Details

- Line 145: Text indicates a different measurement for the short axis of the oval cavity size compared to Fig 2A. Should be ~36Å to match the figure (or figure needs correction)

RE: Thank you for pointing out the discrepancy between the text and Fig. 2a. We have corrected the text to ensure that the dimensions of the oval cavity are consistent with the figure and stated correctly.

- Mutagenesis Strategy:

- Rationale should be provided for using only serines or arginines as substituent amino acids

RE: Thank you for this comment. We have clarified the rationale for our mutagenesis strategy in the manuscript (**Page 10, Lines 224-229**). Specifically, we used serine substitutions to probe the importance of hydrophobic interactions, as serine is a small, polar residue that likely only minimally disrupts the overall structure. In contrast, we used arginine substitutions to assess the impact of introducing a large, positively charged residue at critical interaction sites, which would test the effect of steric hindrance and charge repulsion.

- Several mutations show discrepancies between text and figures, these inconsistencies should be resolved: F461S and L450S appear only in text ; Y456S shown only in Fig 4C.

RE: Thank you for pointing out these inconsistencies. We have now ensured that all mentioned mutations are consistently represented in both the text (**Page 10, Line 233**) and figures.

3. Figure Corrections and Species Comparisons

- Lines 277 and 374: Figure references appear to be incorrect, likely should refer to Fig 1 rather than Fig 2?

RE: Thank you for bringing this to our attention. We have carefully reviewed the figure references throughout the manuscript and corrected the inconsistencies to ensure that all figure citations are accurate and appropriately referenced in the text.

- Fig 6G-H require additional clarification:

- Please elaborate further about the significant difference in FliD heads (D1-D2 angle) between *S. enterica* and *C. jejuni* caps (56° vs 106°)

- Are there potential impacts on the pentamer structures due to this?

- It is not clear to me whether this is related to your paragraph lines 387-395?

- The D1-D2 hinge angle disparity should be visible in Fig 6H's superposition: could you label this better to understand what is happening? I agree with the authors in that these differences may correspond to species-specific traits, yet they are indeed important features to be shown.

RE: Thank you for your feedback on Figure 6 and the respective paragraphs in the manuscript. We have revised the manuscript to clarify the differences in the D1-D2 hinge angle between *S. enterica* and *C. jejuni* FliD caps (now indicated on **Page 20, Lines 415-431**). We initially miscalculated the D1-D2 angle for *C. jejuni*; the correct

value for FliD 3 is 82°, rather than 106°. Thank you for bringing this important correction to our attention.

Our updated analysis confirms that FliD 3 exhibits the most distinct hinge angle in both species, likely reflecting different stages of flagellin incorporation. The 56° angle in *S. enterica* corresponds to an ongoing incorporation cycle, while the 82° angle in *C. jejuni* represents a pre-incorporation state, where the cap is positioned on the HFJ but not actively engaged in flagellin polymerization. Despite the differences, the overall pentamer architecture remains highly similar between the two species. However, the presence of a D4 domain in *C. jejuni*, absent in *S. enterica*, may restrict D1-D2 flexibility, potentially explaining the larger hinge angles observed.

We have updated Fig. 6h to clearly label the corrected D1-D2 angles and improved the visualization of structural differences, including the flat (*C. jejuni*) vs. tilted (*S. enterica*) D2/D3 plane and a superimposition of FliD 3, where the structural differences are clearly labelled.